# A systematic evaluation of single cell RNA-seq analysis pipelines

Beate Vieth [1], Swati Parekh [2], Christoph Ziegenhain[3], Wolfgang Enard [1] & Ines Hellmann [1]*

The recent rapid spread of single cell RNA sequencing (scRNA-seq) methods has created a large variety of experimental and computational pipelines for which best practices have not yet been established. Here, we use simulations based on five scRNA-seq library protocols in combination with nine realistic differential expression (DE) setups to systematically evaluate three mapping, four imputation, seven normalisation and four differential expression testing approaches resulting in ~3000 pipelines, allowing us to also assess interactions among pipeline steps. We find that choices of normalisation and library preparation protocols have the biggest impact on scRNA-seq analyses. Specifically, we find that library preparation determines the ability to detect symmetric expression differences, while normalisation dominates pipeline performance in asymmetric DE-setups. Finally, we illustrate the importance of informed choices by showing that a good scRNA-seq pipeline can have the same impact on detecting a biological signal as quadrupling the sample size.

[1] Anthropology and Human Genomics, Department of Biology II, Ludwig-Maximilians University, Munich, Germany. [2] Max Planck Institute for Biology of Ageing, Cologne, Germany. [3] Department of Cell and Molecular Biology, Karolinska Institutet, SE-171 65, Stockholm, Sweden. *email: hellmann@bio.lmu.de

Many experimental protocols and computational analysis approaches exist for single cell RNA sequencing (scRNA-seq). Furthermore, scRNA-seq analyses can have different goals including differential expression (DE) analysis, clustering of cells, classification of cells and trajectory reconstruction[1]. All these goals have the first analysis steps in common in that they require expression counts or normalised counts. Here, we focus on these important first choices made in any scRNA-seq study, using DE-inference as performance readout. Benchmarking studies exist only separately for each analysis step, which are library preparation protocols[2,3], alignment[4,5], annotations[6], count matrix preprocessing[7,8] and normalisation[9]. However, the impact of the combined choices of the separate analysis steps on overall pipeline performance has not been quantified. In order to achieve a fair and unbiased comparison of computational pipelines, simulations of realistic data sets are necessary. This is because the ground truth of real data is unknown and alternatives, such as concordance analyses are bound to favour similar and not necessarily better methods.

To this end, we integrate popular methods for each analysis step into our simulation framework powsimR[10]. As the basis for simulations, powsimR uses raw count matrices to describe the mean-variance relationship of gene expression measures. This includes the variance introduced during the experiment itself as well as extra variance due to the first to computational steps of expression quantification. Adding DE then provides us with detailed performance measures based on how faithfully DE-genes can be recovered.

One main assumption in traditional DE-analysis is that differences in expression are symmetric. This implies that either a small fraction of genes is DE while the expression of the majority of genes remains constant or similar numbers of genes are up- and down-regulated so that the mean total mRNA content does differ between groups[11]. This assumption is no longer true when diverse cell types are considered. For example, Zeisel et al.[12] find up to 60% DE genes and differing amounts of total mRNA levels between cell types. This issue of asymmetry is conceptually one of the characteristics that distinguishes single cell from bulk RNA-seq and has not been addressed so far. Therefore, we simulate varying numbers of DE-genes in conjunction with small to large differences in mRNA content including the entire spectrum of possible DE-settings.

Realistic simulations in conjunction with a wide array of scRNA-seq methods, allow us not only to quantify the performance of individual pipeline steps, but also to quantify interdependencies among the steps. Moreover, the relative importance of the various steps to the overall pipeline can be estimated. Hence, our analysis provides sound recommendations regarding the construction of an optimal computational scRNA-seq pipeline for the data at hand.

## Results

**scRNA-seq data and simulations**. The starting point for our comprehensive pipeline comparison is a representative selection of scRNA-seq library preparation protocols (Fig. 1a). Here, we included one full-length method (Smart-seq2[13]) and four UMI methods[2,14–16]. The UMI strategies encompass two plate-based (SCRB-seq, CEL-seq2) and the most common non-commercial and commercial droplet-based protocols (Drop-seq, 10X Chromium). CEL-seq2 differs from SCRB-seq in that it relies on linear amplification by in vitro transcription, while SCRB-seq relies on PCR amplification using the same strategy as 10X Chromium (see Ziegenhain et al.[2,17] for a detailed discussion). We then combine the library preparation protocols with three mapping approaches[18–20] and three annotation schemes[21–23] resulting in 45

distinct raw count matrices (see "Methods"). We simulated 27 distinct DE-setups per matrix, each with 20 replicates, resulting in a total of 19,980 simulated data sets (Fig. 1b).

**Genome-mapping quantifies gene expression with high accuracy**. We first investigated how expression quantification is affected by different alignment methods using our selection of scRNA-seq experiments. For each of the three following strategies we picked one the most popular methods (Supplementary Fig. 2): (1) alignment of reads to the genome using splice-aware alignment (STAR[18]), (2) alignment to the transcriptome (BWA[19]) and (3) pseudo-alignment of reads guided by a transcriptome (kallisto[24]).We then combined these with three annotation schemes including two curated schemes (RefSeq[21] and Vega[23]) and the more inclusive GENCODE[22] (Supplementary Table 2).

First, we assessed the performance by the number of reads or UMIs that were aligned and assigned to genes (Fig. 2a and Supplementary Fig. 3). Alignment rates of reads are comparable across all scRNA-seq protocols. Assignment rates on the other hand show some interaction between mapper and protocol. All mappers, aligned and assigned more reads using GENCODE as compared to RefSeq annotation, whereas the pseudo-aligner kallisto profited most from the more comprehensive annotation of GENCODE and here in particular the 3'UMI protocols (Figure 2A). Generally, STAR in combination with GENCODE aligned (82–86%) and assigned (37–63%) the most reads, while kallisto assigned consistently the fewest reads (20-40%) (Figure 2D). BWA assigned an intermediate fraction of reads (22–44%), but—suspiciously—these were distributed across more UMIs. As reads with the same UMI are more likely to originate from the same mRNA molecule and thus the same gene, the average number of genes with which one UMI sequence is associated, can be seen as a measure of false mapping. Indeed, we find that the same UMI is associated with more genes when mapped by BWA than when mapped by STAR (Fig. 2b). This indicates a high false mapping rate, that probably inflates the number of genes that are detected by BWA (Fig. 2c and Supplementary Fig. 4).

This said, it remains to be seen what impact the differences in read or UMI counts obtained through the different alignment strategies and annotations have on the power to detect DE-genes.

As already indicated from the low fraction of assigned reads, kallisto has the lowest mean expression and the highest gene dropout rates (Fig. 2d and Supplementary Figs. 7 and 8) and, as expected from a high fraction of falsely mapped reads, BWA has the largest variance. To estimate the impact that these statistics have on the power to detect DE-genes, we use the mean-variance relationship to simulate data sets with DE-genes (Fig. 2d, e). As previously reported[2], UMI protocols have a noticeably higher power than Smart-seq2 (Fig. 2f). Moreover for Smart-seq2, we find that kallisto, especially with RefSeq annotation, performs slightly better than STAR, while for UMI-methods STAR performs better (Fig. 2f and Supplementary Fig. 9).

In summary, using BWA to map to the transcriptome introduces noise, thus considerably reducing the power to detect DE-genes as compared to genome alignment using STAR or the pseudo-alignment strategy kallisto, but given the lower mapping rate of kallisto STAR with GENCODE is generally preferable.

**Many asymmetric changes pose a problem without spike-ins**. The next step in any RNA-seq analysis is the normalisation of the count matrix. The main idea here is that the resulting size factors correct for differing sequencing depths. In order to improve normalisation, spike-ins as an added standard can help, but are not feasible for all scRNA-seq library preparations. Another avenue to improve normalisation would be to deal with sparsity

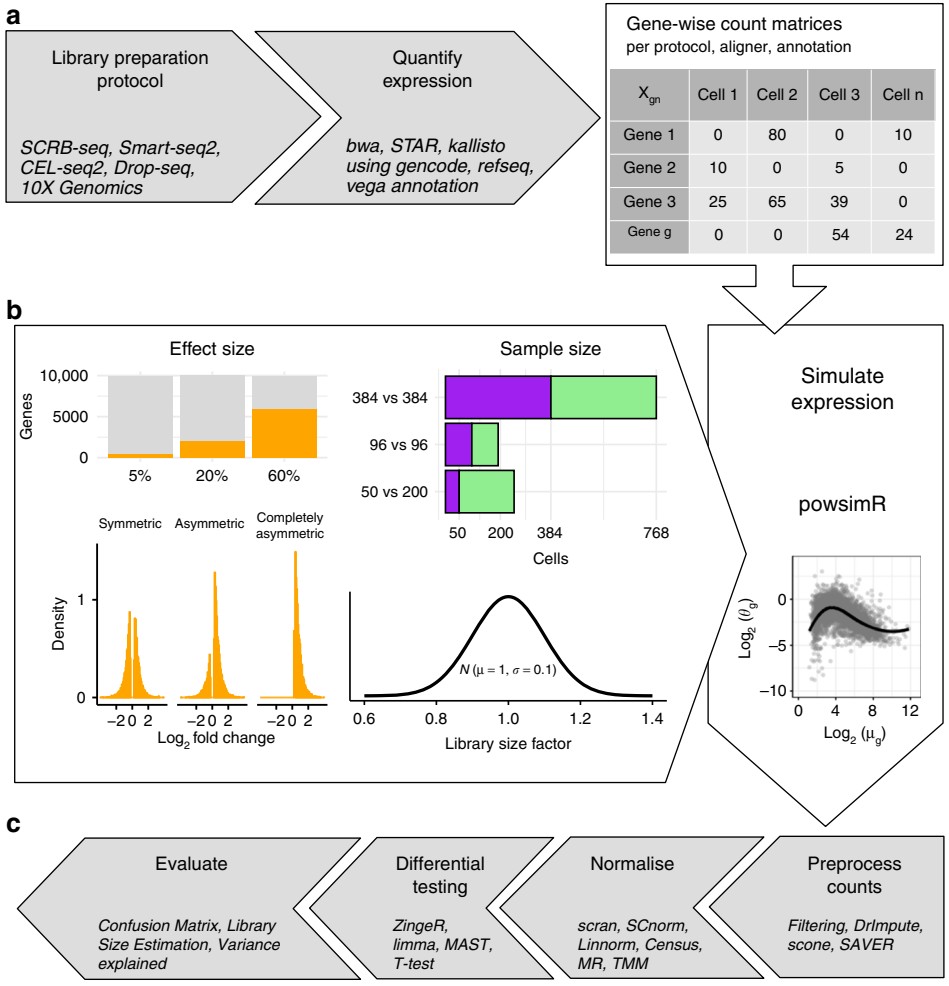

**Fig. 1** Study Overview. **a** The data sets yielding raw count matrices: We use scRNA-seq data sets[2,16] representing 5 popular library preparation protocols. For each data set, we obtain multiple gene count matrices that result from various combinations of alignment methods and annotation schemes (see also Supplementary Figs. 1 and 2, and Supplementary Tables 1 and 2). **b** The simulation setup: Using powsimR[10] distribution estimates from real count matrices, we simulate the expression of 10,000 genes for two groups with 384 vs 384, 96 vs. 96 and 50 vs. 200 cells, where 5, 20 or 60% of genes are DE between groups. The magnitude of expression change for each gene is drawn from a narrow gamma distribution ($X \sim \Gamma(\alpha = 1, \beta = 2)$) and the directions can either be symmetric, asymmetric or completely asymmetric. To introduce slight variation in expression capture, we draw a different size factor for each cell from a narrow normal distribution. **c** The analysis pipeline: The simulated data sets are then analysed using combinations of four count matrix preprocessing, seven normalisation and four DE approaches. The evaluation of these pipelines focuses on the outcome of the confusion matrix and its derivatives (TPR, FDR, pAUC, MCC), deviance in library size estimates (RMSE) and computational run time

by imputing missing data prior to normalisation as discussed in the next chapter (Fig. 1c). To begin with, we compare how much the estimated size factors deviate from the truth. As long as there is only a small proportion of DE-genes or if the differences are symmetric, estimated size factors are not too far from the simulated ones and there are no large differences among methods (Fig. 3a and Supplementary Figs. 10 and 11). However with increasing asymmetry, size factors deviate more and more and the single cell methods scran[25] and SCnorm[26] perform markedly better than the bulk methods TMM[27], MR[28] and Positive Counts as well as the single cell method Linnorm[29]. Census[30] is an outlier in that it has a constant deviation of 0.1, which is due to filling in 1 when library sizes could not be calculated.

To determine the effect of these deviations on downstream analyses, we evaluated the performance of DE inference using different normalisation methods (Fig. 3b and Supplementary Figs. 12–15). Firstly, the differences in the TPR across normalisation methods are only minor, only Linnorm performed consistently worse (Supplementary Fig. 13). In contrast, the ability to control the FDR heavily depends on the normalisation

method (Supplementary Fig. 14). For small numbers of DE-genes or symmetrically distributed changes, the FDR is well controlled for all methods except Linnorm. However, with an increasing number and asymmetry of DE-genes, only SCnorm and scran keep FDR control, provided that cells are grouped or clustered prior to normalisation. In our most extreme scenario with 60% DE-genes and complete asymmetry, all methods except Census loose FDR control. SCnorm, scran, Positive Counts and MR regain FDR control with spike-ins for 60% completely asymmetric DE-genes (Supplementary Fig. 14). Given similar TPR of the methods, this FDR control determines the pAUC (Fig. 3b, c).

Since in real data it is usually unknown what proportion of genes is DE and whether cells contain differing levels of mRNA, we recommend a method that is robust under all tested scenarios. Thus, for most experimental setups scran is a good choice, only for Smart-seq2 data without spike-ins, Census might be a better choice.

**Imputation has little impact on pipeline performance.** If the main reason why normalisation methods perform worse for

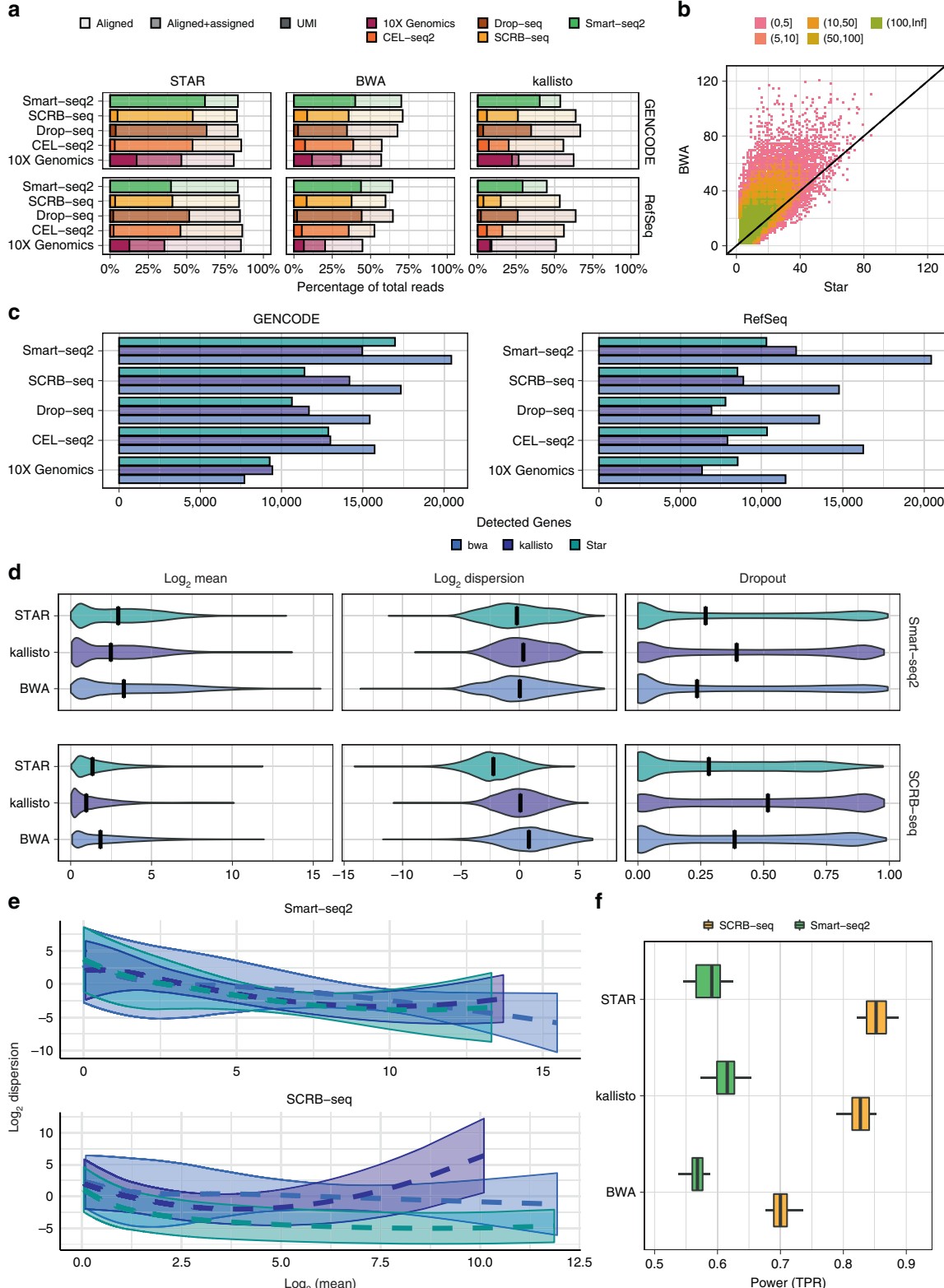

scRNA-seq than for bulk data is the sparsity of the count matrix, reducing this sparsity by either more stringent filtering or imputation of missing values should remedy the problem[31]. Here, we test the impact of frequency filtering and three imputation approaches (DrImpute[32], scone[33], SAVER[34]) on normalisation performance. Note, that we use the imputation or filtering only to

obtain size factor estimates, that are then used together with the raw count matrix for DE-testing.

We find that simple frequency filtering has no effect on normalisation results (Fig. 3d). Performance as measured by pAUC is identical to using raw counts. In contrast, imputation can have an effect on performance and there are large differences

**Fig. 2** Expression Quantification. **a** Read alignment and assignment rates per library preparation protocol stratified over aligner and annotation. The lighter shade represents the percentage of the total reads that could be aligned and the darker shade the percentage that also was uniquely assigned (see also Supplementary Fig. 3). For comparability, cells were downsampled to 1 million reads/cell, with the exception of 10× Genomics data that were only sequenced to on average 60,000 reads/cell. Hence, these data are farther from saturation and have a higher UMI/read ratio. **b** Number of genes per UMI with >1 reads for BWA and STAR alignment using the SCRB-seq data set and GENCODE annotation. Colours denote number bins of UMIs. **c** Number of genes detected per Library Preparation Protocol stratified over Aligner and Annotation (i.e. at least 10% nonzero expression values) (see also Supplementary Fig. 4). **d** Estimated mean expression, dispersion and gene dropout rates for SCRB-seq and Smart-seq2 data using STAR, BWA or kallisto alignments with GENCODE annotation (see also Supplementary Fig. 7). **e** Mean-dispersion fitting line applying a cubic smoothing spline with 95% variability bands for SCRB-seq and Smart-seq2 data using STAR, BWA or kallisto alignments with GENCODE annotation (see also Supplementary Fig. 8). **f** The effect of quantification choices on the power (TPR) to detect differential expression stratified over library preparation and aligner. The expression of 10,000 detected genes over 768 cells (384 cells per group) were simulated given the observed mean-variance relation per protocol. Five percent of the simulated genes are differentially expressed following a symmetric narrow gamma distribution. Unfiltered counts were normalised using scran. Differential expression was tested using limma-trend (see also Supplementary Fig. 9)

among methods. Imputation with DrImpute and scone rarely increased the pAUC and occasionally as in the case of SCRB-seq with MR normalisation, the pAUC even decreased by 100 and 76%, respectively due to worse FDR control relative to using raw counts (Supplementary Fig. 18). In contrast, these two imputation methods achieved an appreciable increase in pAUC together with scran normalisation, ~28, 4 and 9% for 10× Genomics, SCRB-seq and Smart-seq2 data, respectively. SAVER on the other hand never made things worse, irrespective of data set and normalisation method but was able to rescue FDR control for MR normalisation of UMI data, even in a completely asymmetric DE-pattern.

These observations suggest that data sets with a high gene dropout rate might benefit more from imputation than data sets with a relatively low gene dropout rate (Supplementary Figs. 16–18). In order to further investigate the effect of imputation on sparse data, we downsampled the Smart-seq2 and SCRB-seq data, which were originally based on 1 million reads/cell, to make them more comparable to the 10X-HGMM data with on average of 60,000 reads/cell. A radical downsampling to 10% of the original sequencing depth decreases the number of detected genes for SCRB-seq by only 1%, suggesting that the original RNA-seq library was sequenced to saturation. In contrast, the Smart-seq2 data were much less saturated at 1 million reads/cell: Downsampling reduced the number of detected genes by 34%. However, the relative effect of imputation on performance remains small. This is probably due to the fact that the main effect of downsampling is a reduction in the detected genes, which also cannot be imputed. Thus, if a good normalisation method is used to begin with (e.g. scran with clustering), the improvement by imputation remains relatively small.

**Good normalisation removes the need for specialised DE-tools**. The final step in our pipeline analysis is the detection of DE-genes. Recently, Soneson et al.[31] benchmarked 36 DE approaches and found that edgeR[27], MAST[35], limma-trend[36] and even the T-Test performed well. Moreover, they found that for edgeR, it is important to incorporate an estimate of the dropout rate per cell. Therefore, we combine edgeR here with zingeR[37].

Both edgeR-zingeR and limma-trend in combination with a good normalisation reach similar pAUCs as using the simulated size factors (Fig. 4). However, in the case of edgeR-zingeR this performance is achieved by a higher TPR paid while loosing FDR control (see Supplementary Figs. 19–21), even in the case of symmetric DE-settings (Supplementary Figs. 22–24).

Nevertheless, we find that DE-analysis performance strongly depends on the normalisation method and on the library preparation method. In combination with the simulated size factors or scran normalisation, even a T-Test performs well.

Conversely, in combination with MR or SCnorm, the T-Test has an increased FDR (Supplementary Fig. 20). SCnorms bad performance with a T-Test was surprising given SCnorms good performance with limma-trend (Fig. 3b). One explanation could be the relatively large deviation of SCnorm derived size factors (Fig. 3a and Supplementary Fig. 11) which inflate the expression estimates.

Furthermore, we find that MAST performs consistently worse than the other DE-tools when applied to UMI-based data, but -except in combination with SCnorm- it is doing fine with Smart-seq2 data. Interestingly, Census normalisation in combination with edgeR-zingeR outperformed limma-trend with Smart-seq2 (Supplementary Fig. 25).

In concordance with Soneson et al.[31], we found that limma-trend, a DE-tool developed for bulk RNA-seq data showed the most robust performance. Moreover, library preparation and normalisation appeared to have a stronger effect on pipeline performance than the choice of DE-tool.

**Normalisation is overall the most influential step**. Because we tested a nearly exhaustive number of ~3000 possible scRNA-seq pipelines, starting with the choice of library preparation protocol and ending with DE-testing, we can estimate the contribution of each separate step to pipeline performance for our different DE-settings (Fig. 1b). We used a beta regression model to explain the variance in pipeline performance with the choices made at the seven pipeline steps (1) library preparation protocol, (2) spike-in usage, (3) alignment method, (4) annotation scheme, (5) preprocessing of counts, (6) normalisation and (7) DE-tool as explanatory variables. We used the difference in pseudo-$R^2$ between the full model including all seven pipeline steps and leave-one-out reduced models to measure the contribution of each separate step to overall performance.

All pipeline choices together (the full model) explain ~50 and ~60% of the variance in performance, for 20 and 60% DE-genes, respectively (Fig. 5a). Choices of preprocessing the count matrix contribute very little ($\Delta R^2 \leq 1\%$). The same is true for annotation ($\Delta R^2 \leq 2\%$) and aligner choices ($\Delta R^2 \leq 5\%$). For aligner and annotation, it is important to note that these are upper bounds, because our simulations do not include differences in gene detection rates (Fig. 2c).

Surprisingly, the choice of DE-tool only matters for symmetric DE-setups ($\Delta R^2_{\mathrm{DE}=0.2} = 15\%$; $\Delta R^2_{\mathrm{DE}=0.6} = 11\%$), and the choice of library preparation protocol has an even bigger impact on performance for symmetric DE-setups ($\Delta R^2_{\mathrm{Symmetric}} = 17 - 29\%$) and additionally for 5% asymmetric changes ($\Delta R^2_{5\%\mathrm{Asymmetric}} = 17\%$). Normalisation choices have overall a large impact in all DE-settings ($\Delta R^2 = 12–38\%$), where the importance increases with increasing levels of DE-genes and increasing asymmetry. Spike-ins are

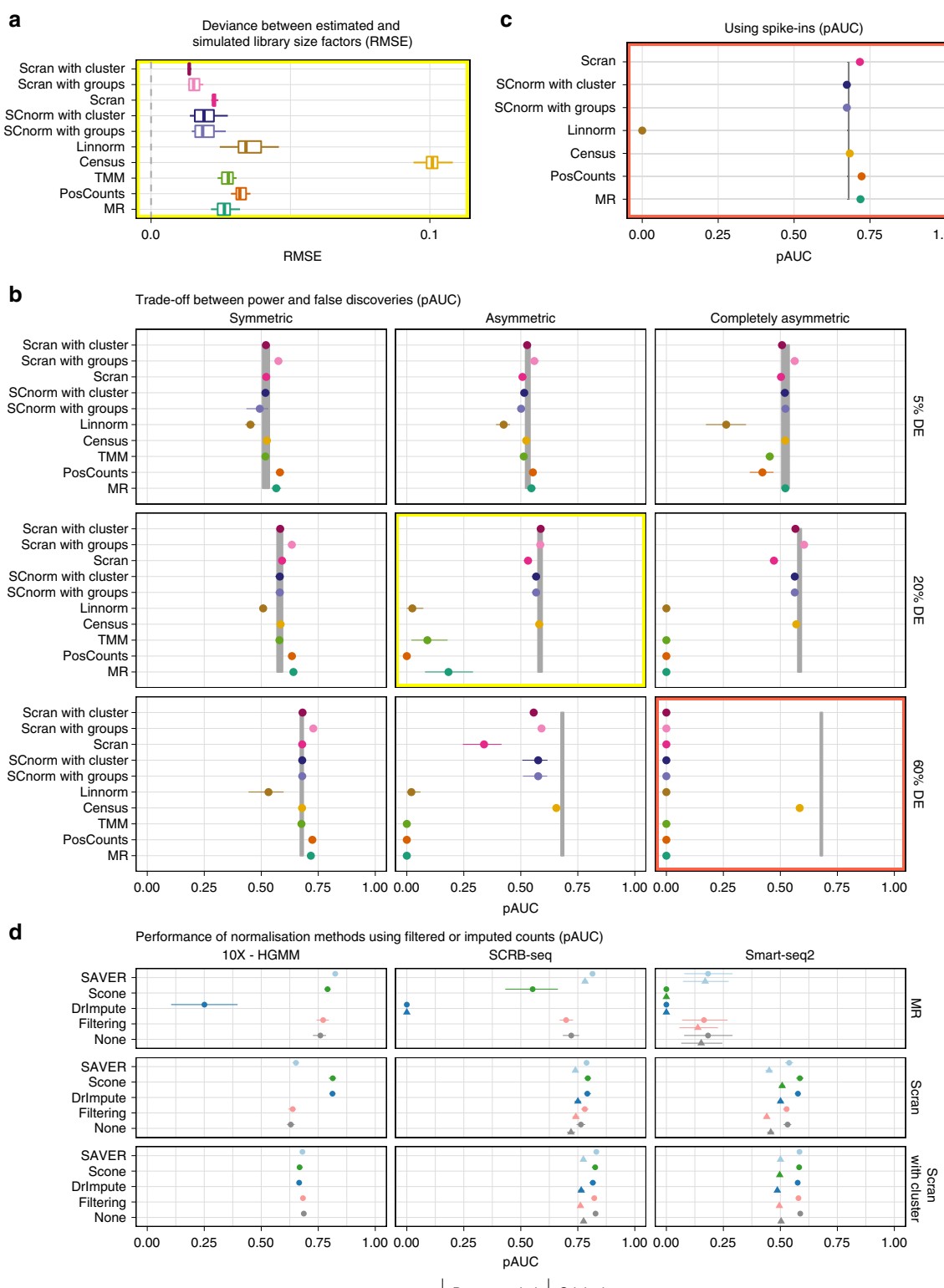

only necessary if many asymmetric changes are expected and have little or no impact if only 5% of the genes are DE or the changes are symmetric (Fig. 5a). Moreover, for completely asymmetric DE-patterns, the regression model did not converge without normalisation and spike-ins, because their absence or presence alone pushed the MCCs to the extremes.

For the best performing pipeline [SCRB-seq + STAR + GENCODE + SAVER imputation + scran with clustering + limma-trend], using 384 cells per group instead of 96 improves performance only by 6.5–8%. Sample size is more important if a naive pipeline is used. For [SCRB-seq + BWA + GENCODE + no count matrix preprocessing + MR + T-Test] the performance gain by increasing sample size is 10–12% and even worse, for many asymmetric DE-genes, lower sample sizes occasionally appear to perform better (Fig. 5b and Supplementary Fig. 26). Next, we tested our pipeline on publicly available 10× Genomics data set containing the expression profiles of approx. 1000 human peripheral mononuclear blood cells

**Fig. 3** Normalisation choices determines DE-analysis performance, not count preprocessing. The data in panels **a–c** are based on Smart-seq2 data, all panels are based on two groups of 384 cells, STAR alignment with GENCODE annotation was used for quantification. **a** The root mean squared error (RMSE) of estimated library size factors per normalisation method is plotted for 20% asymmetric DE-genes (see also Supplementary Fig. 11) (Box and whisker plot with centre line = median, bounds of box = 25th and 75th percentile, whiskers = 1.5 * interquartile range from the lower and upper bounds of the box). **b** The discriminatory ability determined by the partial area under the curve (mean pAUC ± s.d.) based on DE testing with limma-trend for normalisation without spike-ins per DE-pattern. The grey ribbon indicates the mean pAUC ± s.d. given simulated size factors (see also Supplementary Figs. 13–15). **c** Using spike-ins for normalisation for 60% completely asymmetric DE-genes. **d** Effect of preprocessing the count matrix for 20% asymmetric DE-genes without spike-ins. Counts were either left asis ('none'), filtered or imputed prior to normalisation. The derived scaling factors were then used for normalisation and DE testing was performed on raw counts using limma-trend (see also Supplementary Figs. 16–18). This procedure was applied to the full count matrix (circle) and to the count matrix downsampled to 10% of its original sequencing depth (triangular). Missing data points are due to failing imputation runs with the sparser data

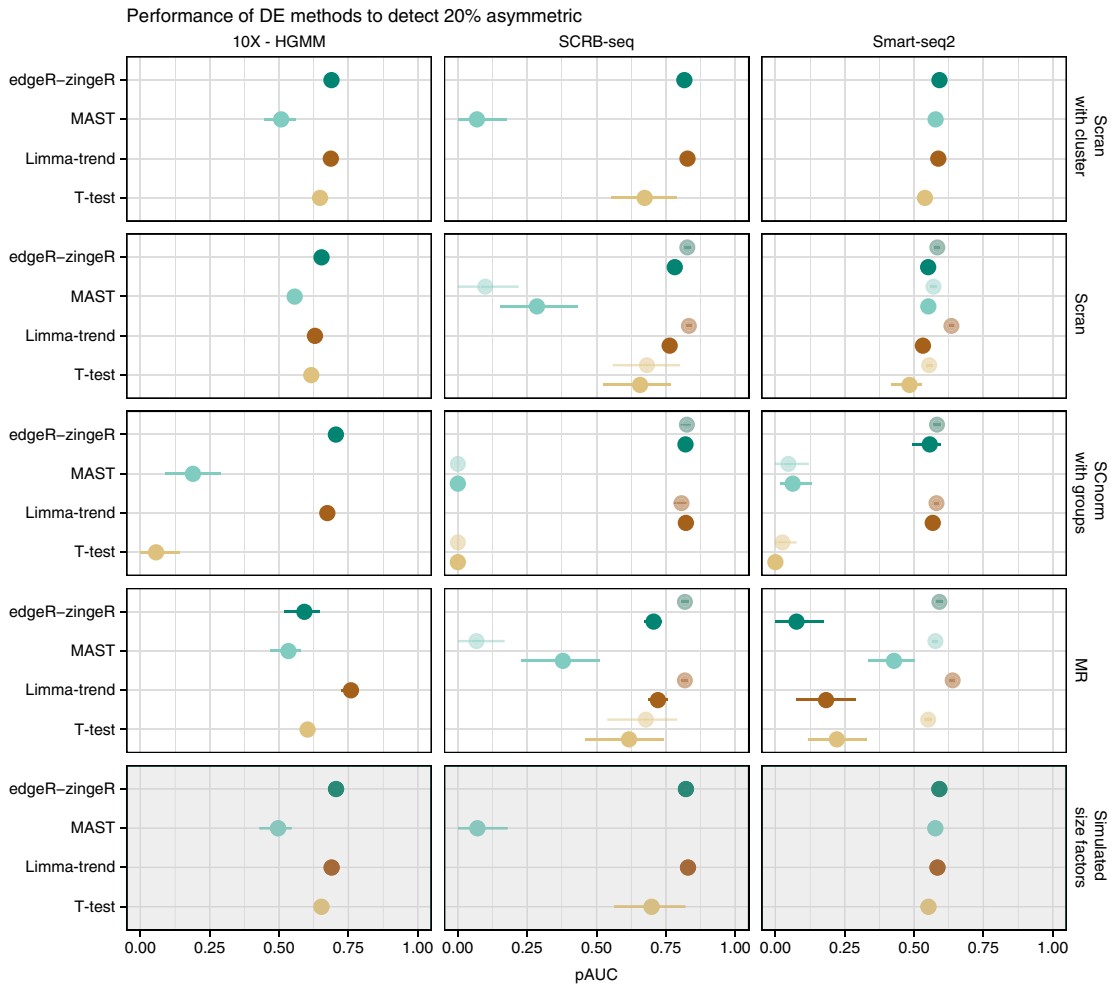

**Fig. 4** Evaluation of DE tools. The expression of 10,000 genes over 768 cells (384 cells per group) were simulated given the observed mean-variance relation per protocol. Twenty percent of the simulated genes are differentially expressed following an asymmetric narrow gamma distribution. Unfiltered counts were normalised using simulated library size factors or applying normalisation methods. Differential expression was tested using *T*-Test, limma-trend, MAST or edgeR-zingeR. The discriminatory ability of DE methods is determined by the partial area under the curve (mean pAUC ± s.d.) for the TPR-FDR curve (see also Supplementary Figs. 19–25)

(PBMC)[16]. First, we classified the cells using SingleR[38] into the celltypes available in the Blueprint Epigenomics Reference[39] distinguishing Monocytes, NK-cells, CD8 + T-cells, CD4 + T-cells and B-cells (Fig. 5c, d). We applied the previously defined good (STAR + gencode + SAVER imputation + scran with clustering + limma-trend) and naive (BWA + gencode + no preprocessing + MR + *T*-Test) pipeline to identify DE-genes between the cell types. Cross-referencing the identified DE-genes with known differences in marker gene expression[39], we find that the good pipeline always identifies a higher fraction of the marker genes as DE than the naive pipeline (Fig. 5e). Comparing NK-cells and CD8 + T-cells, the good pipeline identifies 148 known markers as DE, while the naive pipeline finds only 54. The diminished separation between those two cell-types using the naive pipeline is already visible in the UMAP (Fig. 5d).

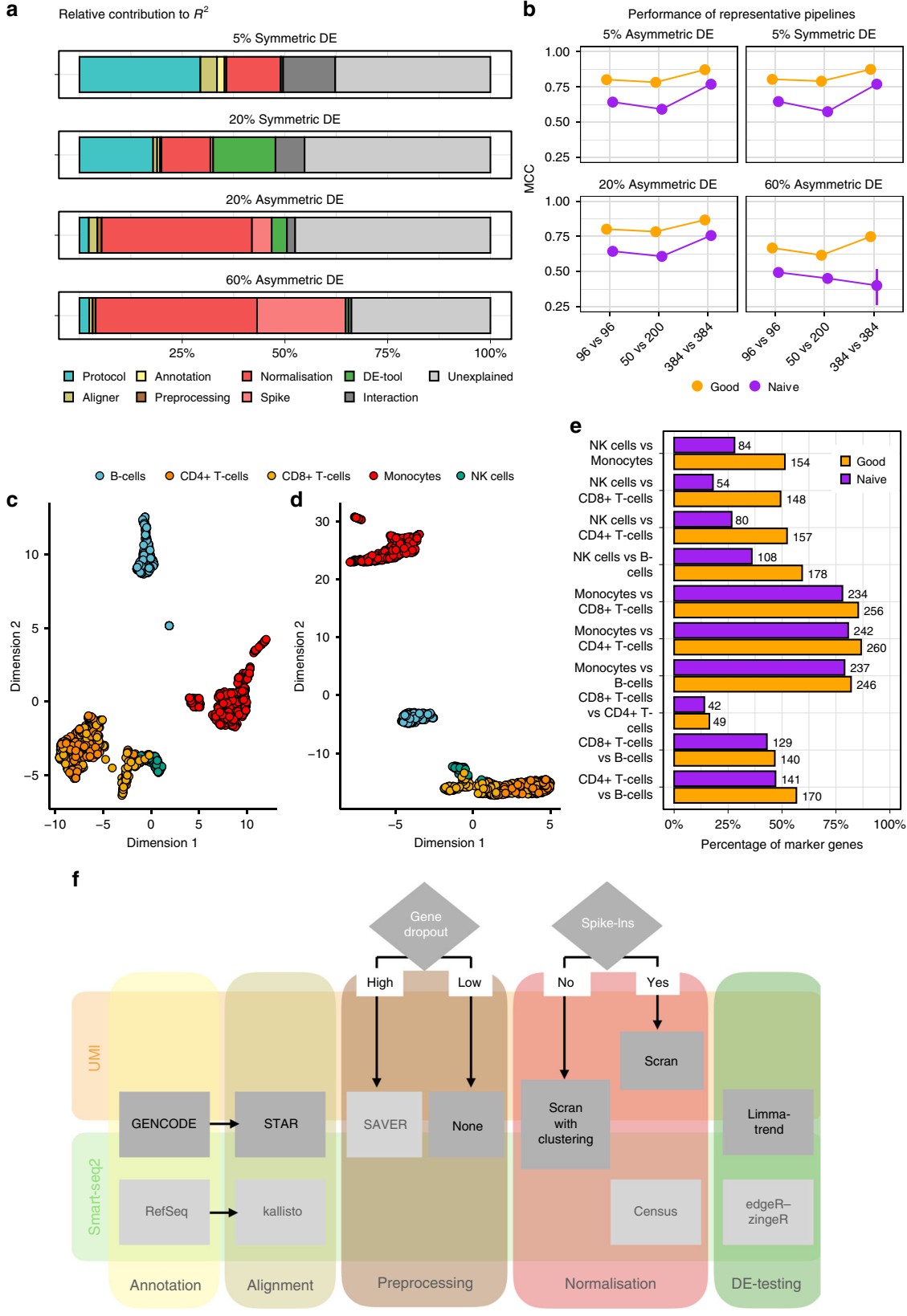

In summary, we identify normalisation and library preparation as the most influential choices and the observation that differences in computational steps alone can significantly lower the required sample size nicely illustrates the importance of bioinformatic choices.

## Discussion

Here we evaluate the performance of complete computational pipelines for the analysis of scRNA-seq data under realistic conditions with large numbers of DE-genes and differences in total mRNA contents between groups (Fig. 1). Furthermore, our

**Fig. 5** Evaluation of analysis pipeline. **a**, **b** The expression of 10000 genes over 768 cells were simulated and 5, 20 or 60% of the genes were differentially expressed following a symmetric or asymmetric narrow gamma distribution. This simulation setup was applied to protocols, alignments, annotations, preprocessing of counts, normalisation and DE tools. For each analysis set, the Matthew Correlation Coefficient (mean MCC ± s.d.) was averaged over 20 simulations and rescaled to [0, 1] interval. The MCC was used as a response variable in beta regression models with log-log link function. **a** The contribution of each covariate in the full model (~Protocol + Aligner + Annotation + Preprocessing + Normalisation + DE-Tool). **b** Performance according to sample size, 1 good and 1 naive pipeline (see also Supplementary Fig. 26). **c–e** The expression of ~1000 human PBMCs were profiled with 10× Genomics were processed using the good and naive pipeline. Cell types were identified with SingleR classification using the Blueprint Epigenomics Reference. Cell types are represented in a UMAP, for good **c** and naive **d** pipeline, respectively. True marker genes, i.e., given by the reference, per pairwise comparison of cell types for the good and naive pipeline are given in **e** where genes needed to have a adjusted p-value < 0.1, absolute log2 fold change threshold (>0.1) and expressed in at least 10% of the cells to be considered. **f** Pipeline recommendations for UMI and Smart-seq2 data

simulations allow us not only to investigate the influence of choices made at each pipeline step separately, but also to estimate the relative importance and interactions between different steps of an entire scRNA-seq analysis pipeline. We implemented all assessed computational methods and more in powsimR, so that users can easily evaluate pipeline performance given their own data and expected DE-settings.

Beginning with the creation of the raw count matrix, we find that transcriptome mapping with BWA[19] appears to recover the largest number of genes. However, many of these are probably due to falsely mapped reads, also increase expression variance which ultimately results in a lower sensitivity (Fig. 2c–f). In contrast, the pseudo-alignment method kallisto[24] appears to assign reads precisely, but looses a lot of reads leading to a lower mean expression. Finally, a genome mapping approach using the splice-aware aligner STAR[18] in conjunction with GENCODE annotation recovers the most reads with the highest accuracy (Fig. 5f).

Concerning the preprocessing of the count matrix, we found in concordance with Andrews et al.[40] that in particular for sparse data such as 10X, SAVER[34] imputation before normalisation improves performance, while filtering genes has no effect with our data sets and combinations of normalisation and DE-testing methods.

The choice that had the largest impact on performance throughout all tested DE-settings is the choice of normalisation method. Only for symmetric changes, the choice of library preparation protocol had a slightly larger impact than normalisation. In line with Evans et al. (2018)[11], we found that normalisation performance of bulk methods and also some of the single cell methods declined with asymmetry (Fig. 3b). In particular, for 60% completely asymmetric DE-genes only Census retained FDR control. Unfortunately, Census is not recommended for the use with UMI-counts. Thus, for UMI-counts and 60% completely asymmetric changes, only the use of spike-ins could restore test performance. In the debate about the usefulness of spike-ins[17,41], we land on the pro side: Our simulations clearly show that spike-ins are useful in DE-testing settings with asymmetric changes which is likely to be a common phenomenon in scRNA-seq data. Due to good performance across DE-settings and its speed (Supplementary Figs. 22 and 27) we would recommend scran with prior clustering as the best choice for normalisation (Fig. 5f).

The choice in DE-testing method, our final pipeline step had relatively little impact on overall pipeline performance. A good normalisation prior to DE-testing alleviates the need for more complex and thus vulnerable methods, such as for example MASTs hurdle model which implicitly assumes that the CPM values were generated from zero inflated negative binomial count distribution. Indeed, we previously showed that also scRNA-seq data fit a negative binomial distribution rather well and that the previously reported zero-inflation in scRNA-seq data is mainly due to amplification noise which is removed in UMI-data[10]. Hence, it is not surprising that in concordance with Soneson

et al.[31], we find that relatively straight forward DE-testing methods adapted from bulk RNA-seq perform well with scRNA-seq data.

Finally, we want to remark that paying attention to the details in a computational pipeline and in particular to normalisation pays off. The effect of using a good pipeline as compared to a naively compiled one has a similar or even greater effect on the potential to detect a biological signal in scRNA-seq data as an increase in cell numbers from 96 to 384 cells per group (Fig. 5b).

## Methods

**Single cell RNA-seq data sets**. The starting point for our comprehensive pipeline comparison is the scRNA-seq library preparation (Fig. 1a). In our comparison, we included the gene expression profiles of mouse embryonic stem cells (mESC) as published in Ziegenhain et al.[2] (Supplementary Fig. 1). We selected four data sets for our comparison: Smart-seq2[13] a well-based full-length scRNA-seq protocol, CEL-seq2[15] a well-based 3′ UMI-protocol using linear amplification, SCRB-seq a well-based 3′ UMI-protocol with PCR amplification[2,42] and Drop-seq[14] a droplet-based 3′ UMI-protocol. In addition, 92 poly-adenylated synthetic RNA transcripts of known concentration designed by the External RNA Control Consortium (ERCCs)[43] were spiked in for all methods except Drop-seq. All raw cDNA sequencing reads were cut to a length of 45 bases and downsampled to one million cDNA reads per cell (Supplementary Table 1 and Supplementary Fig. 1).

Finally, we added a 10X Chromium data set sequencing mouse NIH3T3 cells[16], yielding ~400 good cells with on average ~60,000 reads/cell and another 10X data set analysing ~1000 human peripheral blood mononuclear cells (PBMCs).

These choices of library preparation protocols cover the diversity in current protocols without imposing partiality due to biological differences and technical handling.

**Gene expression quantification**. For genome mapping and quantification of the UMI-data with a splice-aware aligner, we used the zUMIs[44] (v.0.0.3) pipeline with STAR[18] (v.2.5.3a) and the mouse genome (Mus_musculus.GRm38) together with annotation files (gtf) for GENCODE (vM15), Vega (VEGA68) and RefSeq (Release 85) (Supplementary Table 2). zUMIs is a fast and flexible pipeline for processing scRNA-seq data where cell barcode or UMI reads with low sequence quality reads are filtered out prior to UMI collapsing by sequence identity which yields identical count results as UMI-tools[44,45]. For Smart-Seq2 we use the same pipeline settings as in zUMIs, simply omitting the UMI collapsing step (Supplementary Table 3).

For transcriptome alignment, we downloaded transcriptome fasta files corresponding to the annotations listed above. We used BWA[19] (v0.7.12) to align the scRNA-seq reads to these transcriptomes. We only removed reads that aligned equally well to transcripts of different genes as truly multi-mapped. The remaining reads were tallied up per gene. For UMI data, the reads were collapsed per gene by identity, similar to the strategy recommended in zUMIs.

For kallisto[24] (v0.43.1), a transcriptome-guided pseudo-alignment method, we followed the recommended quantification procedure for scRNA-seq data to yield abundance estimates per equivalence class. To be comparable with other alignment methods, the counts per equivalence class were collapsed per gene. The counts of equivalence classes representing multiple genes were filtered out. For SCRB-seq, CEL-seq2, Drop-seq and 10× Genomics libraries, we chose the UMI-aware quantification option. The ERCC spike-in sequences were appended to the genome or transcriptome sequences for quantification.

**Simulations**. We used powsimR to estimate, simulate and evaluate single cell RNA-seq experiments[10]. PowsimR has been independently validated for benchmarking DE-approaches[31] and consistently reproduces the mean-variance relationship and dropout rates of genes of scRNA-seq data (see also Supplementary Fig. 28). The gene expression quantification using three different aligners in combination with three annotations per library preparation protocol produced 45 count matrices. These count matrices are the basis for our estimation in powsimR.

Genes needed at least one read or UMI count in at least one cell to be considered in the estimation for simulation parameters. Since we[10] and others[46,47] have found previously, we assume that UMI counts follow a negative binomial distribution and only Smart-seq2 needs the inclusion of zero-inflation. To simulate spike-in data, we added an implementation of the simulation framework for pure technical variation of spike-ins described in Kim et al.[48] to powsimR. The parameters required for these simulations were estimated from 92 ERCC spike-ins in the SCRB-seq, CEL-seq2 and Smart-seq2 data, respectively[2]. To evaluate the effect of differing sequencing depths, we added a new module to powsimR that estimates the degree of PCR amplification for UMI data. This allows the user to downsample a read count matrix by binomial thinning as implemented in edgeR thinCounts()[27] and then to reconstruct the corresponding UMI count matrix base on the estimated PCR amplification rates.

For a detailed evaluation of the pipelines, we simulated two groups of cells for pairwise comparisons with the following three sample size setups: 96 vs. 96, 384 vs. 384 or 50 vs. 200 cells (Fig. 1b). For simplicity, we kept the number of genes that we simulated constant at 10,000. To introduce slight variation in expression capture, we draw a different size factor for each cell from a narrow normal distribution ($X \sim N(\mu = 1, \sigma = 0.1)$) (Fig. 1b). This distribution fits the considered data sets well, irrespective of the applied library preparation method. Furthermore, the two groups of cells can have 5, 20 or 60% differentially expressed genes. To capture the asymmetry of observed expression differences, we considered three setups of DE-patterns: symmetric (50% up- and 50% downregulated), asymmetric (75% up- and 25% downregulated) or completely asymmetric (100% upregulated). The magnitude of expression change is drawn from a narrow gamma distribution ($X \sim \Gamma(\alpha = 1, \beta = 2)$) defining the log2 fold change, which is then added to the sampled mean expression. The combination of these parameters results in a total of 27 DE-setups that were then applied to the parameter estimates from 37 count matrices to simulate 20 replicates for each setting, producing a total of 19,980 simulated data sets.

These data sets were then analysed by a nearly exhaustive number of combinations of four imputation strategies (scone, SAVER, DrImpute), gene dropout filtering (remove genes with more than 80% zero expression values) together with seven normalisation approaches (TMM, MR, Linnorm, Census, Linnorm, scran, SCnorm) with or without spike-ins, depending on library preparation protocol and method (Fig. 1c). Normalisation factors were then derived as described in Soneson et al.[31] and used in conjunction with the raw count matrices for DE-testing using four representative approaches (T-Test, limma-trend, edgeR-zingeR, MAST). The resulting p-values were corrected for multiple testing with Benjamini-Hochberg FDR and we applied a threshold level of 10% to define positive test results. All these steps were seamlessly implemented into powsimR (github: https://github.com/bvieth/powsimR). In total we analysed 2,979 different RNA-seq pipelines.

**Evaluation metrics**. To evaluate the normalisation results, we determined the root mean squared error (RMSE) of a robust linear model using the difference between estimated and simulated library size factors as response variable in rlm() implemented in R-package MASS[49] (v.7.3–51.1) (Supplementary Fig. 10)[9].

All other measures are based on the final results of an entire scRNA-seq pipeline ending with DE-testing. Knowing the identity of the genes that were simulated to show differing expression levels and the results of the DE-testing, we used a number of metrics related to the confusion matrix tabulating the number of true positives, false positives, true negatives and false negatives. We define the power to detect DE with the TPR (TPR = $\frac{TP}{TP+FN}$). The false discovery rate is defined as FDR = $\frac{FP}{FP+TP}$. We combine these two measures in a TPR versus FDR curve to quantify the trade-off between true and false discoveries in a genome-wide multiple testing setup as advocated by[50]. We then summarise these curves by their partial area under curve (pAUC) of TPR versus observed FDR that still ensures FDR control at the nominal level of 10% (Supplementary Fig. 11). This way of calculating the AUC is ideal for data with relatively high true negative rates as the partial integration does not punish methods that are over-conservative, i.e. that stay way below the nominal FDR.

To summarise the whole confusion matrix in one representative value we chose the Matthews Correlation Coefficient (MCC = $\frac{TP*TN - FP*FN}{\sqrt{(TP + FP)(TP + FN)(TN + FP)(TN + FN)}}$), because it is a balanced measure ensuring a reliable comparison of method performance across all DE-settings[50,51]. As for the pAUC, we calculated the maximal value of MCC where the cutoff still ensured FDR control at the nominal level of 10%.

To quantify the relative contribution of each step in the analysis pipeline, we used the MCC as a response variable in a beta regression model implemented in R-package betareg (v.3.1–1)[52] with each individual pipeline step. Because the MCC assumes the extremes of 0 and 1 in some DE-settings, we applied the recommended transformation, namely MCC$_{transformed}$ = $\frac{MCC*(n-1)+0.5}{n}$, where n is the sample size[53]. The contribution is then given by the difference between the full model pseudo-$R^2$ containing all covariates versus a model leaving one step out at a time. This is then scaled to the total variance explained to give relative $\Delta R^2$ percentages.

**Reporting summary**. Further information on research design is available in the Nature Research Reporting Summary linked to this article.

## Data availability

Any relevant data are available from the authors upon reasonable request. The scRNA-seq data used in this manuscript are all publicly available, and they are summarised in Supplementary Table 1. The SCRB-seq, Smart-seq2, Drop-seq, CEL-seq2 data are available at the Gene Expression Omnibus (GEO) under accession code GSE75790. The HGMM and PBMC data sets are available at 10x Genomics's official website (https://support.10xgenomics.com/single-cell-gene-expression/datasets). The data produced by the analysis in this manuscript is freely available from the following zenodo data repository (https://doi.org/10.5281/zenodo.3364466).

## Code availability

The software and code used are summarised in Supplementary Tables 3 and 4. A compendium containing processing scripts and detailed instructions to reproduce the analysis for this manuscript is freely available from the following GitHub repository (https://github.com/bvieth/scRNA-seq-pipelines).

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

## Acknowledgements

This work was supported by the Deutsche Forschungsgemeinschaft (DFG) through LMUexcellent, SFB1243 (Subproject A14/A15) and DFG grant HE 7669/1-1. C.Z. is recipient of an EMBO long-term fellowship (ALTF 673-2017).

## Author contributions

B.V. and I.H. conceived the study. B.V. prepared and analysed the scRNA-seq data. B.V. implemented and conducted the simulation and evaluation framework. S.P. and C.Z. helped in data processing and power simulations. W.E. and I.H. supervised the work and provided guidance in data analysis. B.V., I.H. and W.E. wrote the manuscript. All authors read and approved the final manuscript.

## Competing interests

The authors declare no competing interests.
