## [Peer Review File · Nature Communications]

Reviewers' Comments:

Reviewer #1:

Remarks to the Author:

The work by Vieth et al. introduces a comparison of existing analysis tools for single cell RNA-seq data processing. The authors systematically test existing algorithms and assemble them in a data-driven approach to a pipeline which is now recommended to be the most suitable pipeline in order, to retrieve the most biological meaningful data from single cell RNA sequencing. The authors conclude that data normalisation and choice of the library preparation protocol have the greatest impact on data quality and illustrate that following their suggested pipelines subsequently reduces the need for bigger sample sizes.

The bioinformatic analysis from my point of view is sound and the authors are to be lauded for their careful assessment of existing tools. However, I have several concerns that limit the publication of this manuscript in the current form in Nature Communications.

Major concerns:

1) The authors test several common normalization methods concluding that for most library preparation methods "scran after clustering" is the most successful method, with "Census" working particularly well on Smart-seq2 data. Normalization on sparse single cell data means that some single cells with better coverage lose information in order, to make them comparable to cells with poorer coverage. What this reviewer would like to see is whether the normalization indeed helps to interpret biological data. To overcome this problem one either has to worsen single cells with better coverage to become comparable to single cell data with lower coverage or, alternatively imputation can be used to increase the coverage of poorer single cell profiles. How do cell types that are very different from each other and cell types that are very closely related but still distinct behave after normalization has been applied to a single cell data set?

2) The authors describe the normalization methods they tested straight after introducing the alignment algorithms. They start with the sentence in line 84 that the next step after alignment is the normalization method. In the next chapter they test imputation methods concluding that imputation has marginal to no beneficial effects on the data. Even in case of sparse single cell data from 10x chromium they do not find a dramatic improvement through imputation. That per se might not be wrong but given the order of steps as they describe them the reader gets the impression that normalization occurs before imputation. If that was indeed the case it doesn't surprise me why imputation had no effect on data quality. In fact their final summary in Figure 5C puts the normalization at the right position after imputation but this does not become clear from the order they present the different pipeline steps.

3) In general I think the authors do a great job from a purely technical analysis perspective. Still this manuscript would profit a lot from having their top pipelines applied to real data to demonstrate the degree of improvement their pipeline brings about in contrast to the original analysis by the group generating and publishing the data. Ideally the best combination of tools should be applied to a real data set of each of the tested wet-lab protocols.

4) In Figure 4 they compare different methods for differential expression analysis. Their final summary in Figure 5C mentions "census" in combination with "edgeR zingR" to be the most successful pipeline for Smart-seq2. However, I cannot find the corresponding analysis that allows this conclusion, neither in the main figures nor in the supplement.

5) In line 173 the authors mention that "preprocessing choices" have very little impact on overall quality. I can't disagree more – the way cells are isolated, stained for FACS, cultured and even the scientist pipetting the NGS workflow have in my experience a substantial impact on data quality and introduce significant batch effects.

6) In line 176 the authors state that their simulation does not include differences in gene detection rates. This is the absolute right thing to do to compare different wet-lab protocols with each other which intrinsically have different sensitivities in detecting gene expression. A comprehensive technical comparison in this regard was published recently by the Teichmann Lab (Svensson et al. 2017, Nature Methods). However, independent of the use of UMIs the authors will need to discuss the pros and cons of each of the wet-lab protocols they tested in order to bring their results into context.

Minor points:

1) The legend in figure 2A is not matching the colours used in the graphs.

Reviewer #2:

Remarks to the Author:

Vieth and colleagues present a systematic evaluation of single-cell RNA-seq analysis pipelines, covering the whole process from raw reads to differential expression (DE) detection between two groups of cells. They evaluate multiple methods for each step, using a range of simulated data sets with various differential expression patterns and underlying data characteristics. They conclude that the choice of normalisation and library preparation methods have the biggest impact on the results of a DE analysis, and that an appropriate choice of pipeline can have the same impact as quadrupling the number of cells.

While I think that it is a commendable effort to attempt to evaluate full scRNA-seq analysis pipelines in order to inform users of how to properly choose among the vast number of available method combinations, in my opinion the manuscript falls short of reaching this goal in several ways, which will be discussed in detail below.

Lack of reproducibility and extendability

(1) The evaluated methods are implemented in the powsimR package, which is available on GitHub. However, it is not clear how that package was called to generate the results shown in the manuscript. In some places, it is indicated that "custom R scripts" were used, but these scripts are not provided. It's also not clear how a reader would go about including an additional method in the evaluation. Without such capabilities, the benchmark will be quickly outdated in this fast-moving field. The authors should provide a code repository with all the necessary code to reproduce the analyses and extend the evaluation to apply additional methods to the same data sets that were used for the analysis.

Limited number of evaluated tools for each step

(2) Despite the large number of methods available for processing of scRNA-seq data, only a subset were selected for inclusion in the evaluation (this is also related to the point above, since I can't see a way to add additional methods to the evaluation). Perhaps most importantly from my point of view, no comparison of quantification methods is done for the 10x Genomics data set, where only a count matrix was downloaded from the 10x Genomics website. Given its popularity in current scRNA-seq studies, and the existence of multiple tools for quantification (e.g., Cell Ranger, alevin, UMItools, umis, kallisto) I think this would have been a highly relevant comparison.

Uncertainty regarding the similarity between the simulated and real data

(3) I'm concerned by the ability of the simulated data to capture all the characteristics of real single-cell RNA-seq data sets, and as a consequence, the generalizability of the conclusions to real

data. Even though the simulations are based on real data matrices, as far as I can judge from the text they represent an idealized situation with only two groups of cells, differing only by a (gene-specific) shift in mean expression. Further, the number of simulated genes appear to be independent of the library preparation protocol, while for real data, the number of detected genes strongly depends on the chosen platform (e.g., 10x vs SmartSeq2). Since the simulated data is not made available, there is also no way for the reader to verify how similar it is to a real data set. At the very least, the authors should show that the simulated data is able to capture all the relevant characteristics of real data sets from the different library preparation platforms.

Unclear/incomplete description of the methodology and the effect on the observations

(4) The different alignment/assignment methods give very different results (in terms of the number of aligned or assigned reads or UMIs), but there is no real attempt to understand whether this is an intrinsic difference between the methods, or an effect of the parameter settings. For example, is there anything in common between the genes detected by one method but not with another (e.g., in terms of sequence uniqueness)?

(5) It was not clear how the kallisto TCCs were converted to transcript/gene counts, and at which point in the process "non-unique" alignments were discarded. Also, the authors write that one explanation for the low assignment rates of kallisto could be that "the 3'end of a gene alone often does not allow to distinguish different transcripts"; however, this is not necessary if only a gene level quantification is required.

(6) What does "BWA alignment with assignment of reads to features" mean? The BWA commands in Table S3 do not contain any code to perform assignment of reads to features. Furthermore, the "bwa samse" command help page states that "Repetitive hits will be randomly chosen". Is this accounted for, or will a read that maps to multiple genes not be detected as such (which would affect the assignment rate, the number of detected genes as well as the variance)?

(7) In ref [2] (Fig. 6), for 64 cells or more, SmartSeq2 showed comparable power to the UMI protocols. Could you elaborate on what causes the low power for SmartSeq2 compared to the other protocols (including 10x Genomics) seen in Figures 2F/S7?

(8) What is happening with the mean-dispersion relationship for the kallisto/SCRB-seq data (Figure S6)? It looks very different from the others (and from what is typically expected for RNA-seq data).

(9) It is not clear whether the authors use 'normalisation factors' and 'size factors' to refer to the same concept. Typically, normalisation factors are used as correction factors and multiplied with the library sizes, while size factors incorporate also the library size differences. This is important, since some of the employed normalisation methods will return size factors, while others return normalisation factors. In addition, in the presence of differential expression, the correct size factors are not necessarily proportional to the library sizes (since normalisation factors are intended to avoid compositionality). It is not completely clear to me which values the authors compare with their RMSE evaluations. Also, it is not clear from the text how the normalisation factors (or size factors) are used in the differential expression evaluation (the text says only "The derived normalisation scaling factors were then used in conjunction with the raw count matrices for DE-testing").

(10) The simulation model (specifically, the introduction of DE genes) is not sufficiently well described. For example, were effect sizes chosen independently of the expression level, and was the dropout probability adapted to the "new" expression level (after the introduction of differential expression)? Also, were the data simulated with different library sizes and/or size factors between cells (if so, what was the range, and was it set to be similar to real data)?

(11) How were the TPR vs FDR curves made in order to estimate the pAUC? With FDR on the x-axis (and TPR on the y-axis), there is no guarantee that this curve is a valid function curve, and thus the area under the curve is not readily interpretable.

Various comments

(12) Why were all raw sequencing reads cut and downsampled to 45 base long, 1M cDNA reads per cell?

(13) It could be made clearer that the alignment/assignment evaluation was done on real data, not the simulated ones.

(14) The statement in the Introduction that "All these goals have the first analysis steps leading to a filtered and normalised count matrix in common" is not quite true - many DE methods (and also increasingly methods for other types of analyses) work on the unnormalised counts and build library size normalisation etc. into the model (so a "normalised" matrix is never generated).

(15) MAST does not assume a zero-inflated negative binomial distribution, but a zero-inflated normal distribution of the log-transformed TPMs/CPMs.

(16) The statement "in concordance with Andrews and Hemberg that in particular for sparse data such as 10X, SAVER imputation before normalisation improves performance" seems a bit overstated, given that Andrews and Hemberg's recommendation is that "SAVER is generally the safest method to use, but that all statistical tests, be the gene-gene correlations, cluster-specific marker genes, or differential expression, should be applied to un-imputed data." Also, it is important to note that SAVER is based on a negative binomial assumption, which as far as I can see was also the model used to generate the simulated data (in both studies).

(17) It should be made clearer early in the paper that the focus is on detection of differential gene expression between groups of cells. "Single-cell RNA-seq analysis pipelines" can refer to many other things, including trajectory inference, clustering, visualization and other types of differential analysis (e.g., differential abundance of cell types, or comparison of gene expression levels between samples from different conditions).

(18) How was the "frequency filtering" done?

(19) I would suggest that the authors give a precise definition of what they mean by the term "dropout rate" (i.e., total fraction of zeros, or zeros not explainable by the count model). For example, zingeR does not estimate the fraction of zeros, but a posterior probability of zeros coming from the count model.

We thank the two Reviewers for their many helpful suggestions and thoughtful comments. We worked hard to address all your concerns and the result is this much improved manuscript:

1. We added a downsampling function to our simulator powsimR, that now allows us to evaluate different sequencing depths and thus improves the comparability between the 10X Chromium and the other library preparation methods.
2. We now added the analysis of a real dataset and show that pipeline choice indeed has an effect on identification and characterization of cell-types in scRNA-seq datasets.
3. We now analyse the 10x data using all combinations of mappers and annotations and provide an analysis into the detection biases that lead to the observed differences in the genes found by the different mappers and annotations.
4. Finally, we worked on the text to clear up some misunderstandings, and provide all our scripts in github (<https://github.com/bvieth/scRNA-seq-pipelines>) to insure full reproducibility.

Please find detailed responses (green font) to all points below.

Reviewer #1 (Remarks to the Author):

Major concerns:

1) The authors test several common normalization methods concluding that for most library preparation methods “scran after clustering” is the most successful method, with “Census” working particularly well on Smart-seq2 data. Normalization on sparse single cell data means that some single cells with better coverage lose information in order to make them comparable to cells with poorer coverage. What this reviewer would like to see is whether the normalization indeed helps to interpret biological data. To overcome this problem one either has to worsen single cells with better coverage to become

comparable to single cell data with lower coverage or, alternatively imputation can be used to increase the coverage of poorer single cell profiles. How do cell types that are very different from each other and cell types that are very closely related but still distinct behave after normalization has been applied to a single cell data set?

In order to illustrate that a good scRNA-seq pipeline indeed helps to interpret the biology behind the data, we now show this by analysing a PBMC data set (see answer to point 3 and reviewer 2 point 3 below, and also in the manuscript page 10, line 228 and new Figure 5 C-E).

Furthermore, we now simulate different sequencing depths: the full data and data downsampled to 10% of the original read number. This makes the SCRB-seq and Smart-seq2 data with 1 million reads/cell more comparable to 10x data with 50-100,000 reads/cells. We find that SCRB-seq UMIs were already sequenced to saturation: 1) Only 1% fewer genes were detected after this downsampling. 2) Performance after downsampling drops by at most 5% (Figure 3D). In contrast, Smart-seq2 data are less saturated at 1 million reads/cell: we detect 34% fewer genes and the performance after downsampling drops by up to 10%. Even though this is interesting, it has little effect on our overall conclusions, which focus on the computational pipeline. We mainly included the different library preparation protocols for reference and to evaluate the applicability of the pipelines. In this setting, we view the sequencing depth as part of the wet lab library preparation and we are mainly seeking to evaluate pipeline choices for a given RNA-seq library preparation protocol.

We added the following text to the manuscript: (Page 8, Line 154)

“In order to further investigate the effect of imputation on sparse data, we downsampled the Smart-seq2 and SCRB-seq data, which were originally based on 1 million reads/cell, to make them more comparable to the 10x-HGMM data with on average of 60,000 reads/cell. A radical downsampling to 10% of the original sequencing depth decreases the number of detected genes for SCRB-seq by only 1%, suggesting that the original RNA-seq library was sequenced to saturation. In contrast, the Smart-seq2 data were much less

saturated at 1 million reads/cell: Downsampling reduced the number of detected genes by 34%. However, the relative effect of imputation on performance remains small. This is probably due to the fact that the main effect of downsampling is a reduction in the detected genes, which also cannot be imputed.”

With respect to the last point on similar and very different cell types, we believe that those scenarios are already present in our simulations: highly differentiated cell types would correspond to 60% DE genes with varying levels of asymmetry, while very similar cell types would correspond to 5% DE genes with symmetry.

Figure 3. Normalisation choices determines DE-analysis performance, not preprocessing of counts.

2) The authors describe the normalization methods they tested straight after introducing the alignment algorithms. They start with the sentence in line 84 that the next step after alignment is the normalization method. In the next chapter they test imputation methods concluding that imputation has marginal to no beneficial effects on the data. Even in case of sparse single cell data from 10x chromium they do not find a dramatic improvement through imputation. That per se might not be wrong but given the order of steps as they describe them the reader gets the impression that normalization occurs before imputation. If that was indeed the case it doesn't surprise me

why imputation had no effect on data quality. In fact their final summary in Figure 5C puts the normalization at the right position after imputation but this does not become clear from the order they present the different pipeline steps.

As the reviewer noted, we were also struggling how to best order the investigated analysis steps. We decided to discuss normalisation first, because - unlike imputation - it is not optional. We now extended the introduction to make this more clear.

Page 6, Line 100: "The next step in any RNA-seq analysis is the normalisation of the count matrix. The main idea here is that the resulting normalisation factors correct for differing sequencing depths. In order to improve normalisation, spike-ins as an externally added standard can help, but are not feasible for all scRNA-seq methods. Another avenue to improve normalisation would be to deal with sparsity by imputing missing data prior to normalisation as discussed in the next chapter (Figure 1C)."

Page 6, Line 137: "Note, that we use the imputation or filtering only to obtain size factor estimates, that are then used together with the raw count matrix for DE-testing."

3) In general I think the authors do a great job from a purely technical analysis perspective. Still this manuscript would profit a lot from having their top pipelines applied to real data to demonstrate the degree of improvement their pipeline brings about in contrast to the original analysis by the group generating and publishing the data. Ideally the best combination of tools should be applied to a real data set of each of the tested wet-lab protocols.

We thank the reviewer for this great suggestion. We decided to apply our recommended and our "naive" pipeline to a single cell RNA-seq dataset from Peripheral Blood Mononuclear Cells (PBMCs). PBMCs have been well studied at various levels, so that there are reliable expression signatures for classification (e.g. Blueprint Epigenomics) and well characterized marker genes for the cell types available.

We have downloaded the publicly available data set of 1000 human PBMC from 10X Genomics and processed it once with our recommended and naive pipeline using cell types classified by SingleR. Indeed, we can show that our recommended pipeline already improves the ability to detect genes, classify cells and especially correctly identify marker genes (new Figure 5 C-E).

We added the following text to the manuscript: (page 10, line 228)

“Next, we tested our pipeline on publicly available 10X Genomics data set containing the expression profiles of approx. 1000 human peripheral mononuclear blood cells (PBMC). First, we classified the cells using SingleR into the cell types available in the Blueprint Epigenomics Reference distinguishing Monocytes, NK-cells, CD8+T-cells, CD4+T-cells and B-cells (Figure 5C,D). We applied the previously defined good (*STAR + gencode + SAVER imputation + scran with clustering + limma-trend*) and naive (*BWA + gencode + no preprocessing + MR + T-Test*) pipeline to identify DE-genes between the cell types. Cross-referencing the identified DE-genes with known differences in marker gene expression, we find that the good pipeline always identifies a higher fraction of the marker genes as DE than the naive pipeline (Figure 5E). Comparing NK-cells and CD8+ T-cells, the good pipeline identifies 148 known markers as DE, while the naive pipeline finds only 54. The diminished separation between those two cell-types using the naive pipeline is already visible in the UMAP (Figure 5D).”

Figure 5. Evaluation of analysis pipeline.

C, D, E The expression of ~ 1000 human PBMCs profiled with 10X Genomics were processed using the good and naive pipeline. Cell types were identified with SingleR classification using the Blueprint Epigenomics Reference. Cell types are represented in a UMAP, for good **C** and naive **D** pipeline, respectively. True marker genes, i.e. given by the reference, per pairwise comparison of cell types for the good and naive pipeline are given in **E** where genes needed to have a adjusted p-value < 0.1 , absolute log2 fold change threshold (> 0.1) and expressed in at least 10% of the cells to be considered.

4) In Figure 4 they compare different methods for differential expression analysis. Their final summary in Figure 5C mentions “census” in combination with “edgeR zingeR” to be the most successful pipeline for Smart-seq2. However, I cannot find the corresponding analysis that allows this conclusion, neither in the main figures nor in the supplement.

We are sorry for the mis-understanding: There were no differences in the interaction between normalisation and the DE-tools edgeR-zingeR and limma-trend. The ordering in Figure 5F (previously Figure 5C) at the bottom simply represents a viable alternative for Smart-seq2 data, while the middle block indicates a pipeline that will work well for both UMI-methods and Smart-seq2.

We have included Census now in Supplementary Figure S25 and added the following text (page 9 line 188): “Interestingly, Census normalisation in combination with edgeR-zingeR outperformed limma-trend with Smart-seq2 (Supplementary Figure S25).”

Figure S25: Performance of DE-tools using Census normalisation.

The expression of 10,000 genes over 768 cells (384 cells per group) were simulated given the observed mean-variance relation of Smart-seq2 data. 20% of the simulated genes are differentially expressed following an asymmetric narrow gamma distribution. Unfiltered counts were normalised using Census method. Differential expression was tested using T-Test, limma-trend, MAST or edgeR-zingeR. The lighter shade indicates the usage of spike-ins for normalisation.

A) The discriminatory ability determined by the partial area under the curve (pAUC) based on the TPR-FDR curve is plotted. B) FDR control. The dashed line indicates the nominal FDR level of 10%. C) The power (TPR) to detect differential expression.

5) In line 173 the authors mention that “preprocessing choices” have very little impact on overall quality. I can’t disagree more – the way cells are isolated, stained for FACS, cultured and even the scientist pipetting the NGS workflow have in my experience a substantial impact on data quality and introduce significant batch effects.

We are very sorry for this confusion, we meant computational preprocessing of the count matrix, i.e. filtering of cells and genes and imputation of gene expression. There is no doubt that any experiment is super sensitive to the experimental preprocessings listed by the reviewer. We now changed the text to spell out more clearly that we mean the preprocessing of the count matrix. A number of examples in the main text:

Page 1, Line 8: “As of now, benchmarking studies exist only separately for each analysis step, which are library preparation protocols, alignment, annotations, **count matrix preprocessing** and normalisation.”

Page 9, Line 201: “We used a beta regression model to explain the variance in pipeline performance with the choices made at the seven pipeline steps 1) library preparation

protocol, 2) spike-in usage, 3) alignment method, 4) annotation scheme, 5) **preprocessing of counts**, 6) normalisation and 7) DE-tool as explanatory variables.”

Page 10, Line 206: “Choices of **preprocessing the count matrix** contribute very little ($\Delta R^2 \leq 1\%$).”

6) In line 176 the authors state that their simulation does not include differences in gene detection rates. This is the absolute right thing to do to compare different wet-lab protocols with each other which intrinsically have different sensitivities in detecting gene expression. A comprehensive technical comparison in this regard was published recently by the Teichmann Lab (Svensson et al. 2017, Nature Methods). However, independent of the use of UMIs the authors will need to discuss the pros and cons of each of the wet-lab protocols they tested in order to bring their results into context.

We agree that wet-lab methods differ in their sensitivity to detect genes. However, the study by the Teichmann Lab is unsuitable for comparing this parameter, as their analysis was restricted to the 92 ERCC spike-ins. Fortunately, Ziegenhain et al. 2017 provides a detailed discussion of the pros and cons of wet-lab protocols CEL-seq2, Drop-seq, Smart-seq C1, Smart-seq2, MARS-seq and SCR-seq. Since we are using the same data as main input for our simulation study, we feel that re-iterating the inherent trade-offs already described in Ziegenhain et al. 2017 would be redundant. Therefore we kindly refer the reviewer to this paper.

We added the following text to the main text: (Page 2, Line 42)

“The UMI strategies encompass two plate-based (SCR-seq, CEL-seq2) and the most common non-commercial and commercial droplet-based protocols (Drop-seq, 10X Chromium). CEL-seq2 differs from SCR-seq in that it relies on linear amplification by *in vitro* transcription, while SCR-seq relies on PCR amplification using the same strategy as 10X Chromium (see Ziegenhain et al. 2017 and 2018 for a detailed discussion). We then combine the library preparation protocols with three mapping approaches and three annotation schemes resulting in 45 distinct raw count matrices (Online Methods). We

simulated 27 distinct DE-setups per matrix, each with 20 replicates, resulting in a total of 19,980 simulated data sets (Figure 1B).”

Minor points:

- 1) The legend in figure 2A is not matching the colours used in the graphs.
We have changed the legend accordingly.

Reviewer #2 (Remarks to the Author):

Lack of reproducibility and extendability

(1) The evaluated methods are implemented in the powsimR package, which is available on GitHub. However, it is not clear how that package was called to generate the results shown in the manuscript. In some places, it is indicated that "custom R scripts" were used, but these scripts are not provided. It's also not clear how a reader would go about including an additional method in the evaluation. Without such capabilities, the benchmark will be quickly outdated in this fast-moving field. The authors should provide a code repository with all the necessary code to reproduce the analyses and extend the evaluation to apply additional methods to the same data sets that were used for the analysis.

We added all the scripts used to produce the evaluations presented here to a github http repository accompanying this paper. Now it should be easy for anybody to reproduce and extend our analyses.

We have added an appropriate section to the Materials and Methods (page 15 line 412 "Code availability -- A compendium containing processing scripts and detailed instructions to reproduce the analysis for this manuscript is available from the following GitHub repository (<https://github.com/bvieth/scRNA-seq-pipelines>)."

Limited number of evaluated tools for each step

(2) Despite the large number of methods available for processing of scRNA-seq data, only a subset were selected for inclusion in the evaluation (this is also related to the point above, since I can't see a way to add additional methods to the evaluation). Perhaps most importantly from my point of view, no comparison of quantification methods is done for the 10x Genomics data set, where only a count matrix was downloaded from the 10x Genomics website. Given its popularity in current scRNA-seq studies, and the existence of

multiple tools for quantification (e.g., CellRanger, alevin, UMItools, umis, kallisto) I think this would have been a highly relevant comparison.

We thank the reviewer for the suggestions. We have chosen our initial setup to represent different flavors of alignment methods and chose representative tools/popular pipelines to quantify gene expression. This paper is by no means a comprehensive comparison of mappers, for this we would kindly refer the reviewer to Baruzzo et al. 2017. For example, Alevin was published one week after we submitted this manuscript. Therefore we chose kallisto as the representative pseudoalignment method.

However, we agree that it is important to provide a more comprehensive evaluation of mapping methods for 10x data, which we now included (Figure 2; Supplementary Figure S3, S4, S7, S8). Note, that the 10x datasets that were available to us differ from the other evaluated library preparation methods in two important properties: 1. They represent very different cells, and thus differing transcriptome complexity and differing intercellular variances are expected. Therefore, a quantitative comparison in the number of detected genes is likely to be unfair. 2. Sequencing of the 10x datasets was much shallower with on average 60,000 reads/cell. While this leads to less saturation, i.e. to a relatively high UMI/read ratio, the 10x data behave like other 3' UMI library methods for all other aspects (Figure 2).

We added the following text to the manuscript:

“Alignment rates of reads are comparable across all scRNA-seq protocols. Assignment rates on the other hand show some interaction between mapper and protocol. Notably, kallisto appears to have more problems assigning reads from the 3' UMI protocols and does fine with the full length protocol Smart-seq2 (Supplementary Figure S5 and S6).”
(page 4, line 62)

“Genes that are specifically missed by kallisto have on average more exons and are associated with more transcripts. It is also the higher number transcripts annotated in Gencode as compared to RefSeq that allows kallisto to perform better with RefSeq than

with Gencode annotation (Supplementary Table S2 and Supplementary Figures S5 and S6).” (page 4, line 77)

We also agree that cellranger is a very popular pipeline, however cellranger was designed to analyse 10X Chromium data and is limited by the read design and the proprietary adapter sequences used by this company. This said, the important parts of cellranger are the same as in zUMIs: STAR + quality based UMI and barcode filtering and final UMI collapsing by identity.

We did not include a comparison of UMI-collapsing methods such as umis and UMI-tools. Umis was specifically designed for pseudoalignment methods and it is unclear in how far it is compatible with other mapping strategies. Furthermore, we compared UMI-tools to basic collapsing after various quality filtering thresholds and found that there were no noteworthy differences already in Parekh et al. 2018. We incorporate this now on page 13, line 319: “zUMIs is a fast and flexible pipeline for processing scRNA-seq data where cell barcode or UMI reads with low sequence quality reads are filtered out prior to UMI collapsing by sequence identity which yields identical count results as UMI-tools.”

Figure S5: Properties of Genes detected by the different mapping strategies.

Mappability is represented as $10^{M^{25}}$, where M^{25} is the lower quartile of the mappability scores¹⁰ across the gene. All other gene properties were extracted from the corresponding annotation file. Genes detected by all three mappers tend to have higher expression levels and a slightly higher mappability. The only other consistent pattern is that genes that escape detection by kallisto have more transcripts and exons.

Figure S6: Differences in genes detected with RefSeq and Gencode annotation.

We used the matching between RefSeq and Gencode transcript annotations that is provided by Gencode and then summarise detection at the gene level. Comparing the genes found using Gencode vs. RefSeq annotation, we find that for the 3-prime method SCR-seq both annotations yield approximately the same number of genes, but $\sim 1,000$ appear annotation-specific. The Gencode annotation is more comprehensive, in that it contains more and often longer transcripts and indeed the Gencode-specific genes are longer and thus 3 mapping reads are easily lost. The RefSeq specific transcripts are more puzzling and the only distinguishing feature that we see is that they appear more GC-rich. For full-length data generated with Smart-seq2, Gencode detects 2,500 more genes than RefSeq and there are almost no RefSeq specific genes. Genes only detected with Gencode are longer and have on average more exons and transcripts.

Uncertainty regarding the similarity between the simulated and real data

(3) I'm concerned by the ability of the simulated data to capture all the characteristics of real single-cell RNA-seq data sets, and as a consequence, the generalizability of the conclusions to real data. Even though the simulations are based on real data matrices, as far as I can judge from the text they represent an idealized situation with only two groups of cells, differing only by a (gene-specific) shift in mean expression. Further, the number of simulated genes appear to be independent of the library preparation protocol, while for real data, the number of detected genes strongly depends on the chosen platform (e.g., 10x vs SmartSeq2). Since the simulated data is not made available, there is also no way for

the reader to verify how similar it is to a real data set. At the very least, the authors should show that the simulated data is able to capture all the relevant characteristics of real data sets from the different library preparation platforms.

To show the validity of our simulations within this paper we have now added a supplementary figure comparing observed expression patterns to our simulations (Supplementary Figure S28 and below) and reference this figure in the Methods section (page 14, line 342). In general, the validity is extensively analysed and shown in Vieth et al. 2017 and Ziegenhain et al. 2017 and as in an independent comparison in Sonesson et al. 2017 (countsimQC).

As laid out in detail in the response to reviewer 1 above, we have now also tested conclusions from our simulations on real data by comparing a recommended and a naive scRNA-seq pipeline to a 10x scRNA-seq data set of approximately 1,000 PBMCs that encompasses at least 5 different cell types. We show that the good pipeline can identify many more of the known marker genes than the naive pipeline, demonstrating the usefulness of our simulations in a real-life scenario.

Figure S28: Comparison of simulated and observed parameters.

The marginal distribution of the observed and simulated \log_2 mean, \log_2 dispersion and gene dropout rate for Smart-seq2, SCR-seq and 10X Genomics HGMM scRNA-seq data. For Smart-seq2 the mean and dispersion value excluding zeroes is plotted since a ZINB distribution is assumed.

Unclear/incomplete description of the methodology and the effect on the observations

(4) The different alignment/assignment methods give very different results (in terms of the number of aligned or assigned reads or UMIs), but there is no real attempt to understand whether this is an intrinsic difference between the methods, or an effect of the parameter settings. For example, is there anything in common between the genes detected by one method but not with another (e.g., in terms of sequence uniqueness)?

We now investigate the properties of the genes that show mapper or annotation effects in Supplementary Figures S5 and S6, respectively. The vast majority of genes has near perfect mappability and there are no discernable differences in mappability between genes that were detected by all three mappers as compared to genes detected by only one or two. It also makes sense that genes that were consistently found, appear to have higher expression levels. However, it is notable that kallisto appears to have more problems with complex genes: Genes specifically missed by kallisto have more exons and more annotated transcripts.

Comparing the genes found using Gencode vs. RefSeq annotation, we find that for the 3' method SCRB-seq both annotations yield approximately the same number of genes, but ~1,000 appear annotation-specific. The Gencode annotation is more comprehensive, in that it contains more and often longer transcripts and indeed the Gencode-specific genes are longer and thus 3' mapping reads are easily lost. The RefSeq specific transcripts are more puzzling and the only distinguishing feature that we see is that they appear more GC-rich. For full-length data generated with Smart-seq2, Gencode detects 2,500 more genes than RefSeq and there are almost no RefSeq specific genes. Genes detected only with Gencode are longer and have on average more exons and transcripts.

In summary, we think we have added an analysis that reasonably explains the observed mapping and annotation differences and have added the following text to the Results: "Alignment rates of reads are comparable across all scRNA-seq protocols. Assignment rates on the other hand show some interaction between mapper and protocol. Notably,

kallisto appears to have more problems assigning reads from the 3'UMI-protocols and does fine with the full length protocol Smart-seq2 (Supplementary Figure S5 and S6).”
 (page 4, line 62)

Figure S5: Properties of Genes detected by the different mapping strategies.

Mappability is represented as $10^{M^{25}}$, where M^{25} is the lower quartile of the mappability scores¹⁰ across the gene. All other gene properties were extracted from the corresponding annotation file. Genes detected by all three mappers tend to have higher expression levels and a slightly higher mappability. The only other consistent pattern is that genes that escape detection by kallisto have more transcripts and exons.

Figure S6: Differences in genes detected with RefSeq and Gencode annotation.

We used the matching between RefSeq and Gencode transcript annotations that is provided by Gencode and then summarise detection at the gene level. Comparing the genes found using Gencode vs. RefSeq annotation, we find that for the 3-prime method SCRBS-seq both annotations yield approximately the same number of genes, but ~1,000 appear annotation-specific. The Gencode annotation is more comprehensive, in that it contains more and often longer transcripts and indeed the Gencode-specific genes are longer and thus 3 mapping reads are easily lost. The RefSeq specific transcripts are more puzzling and the only distinguishing feature that we see is that they appear more GC-rich. For full-length data generated with Smart-seq2, Gencode detects 2,500 more genes than RefSeq and there are almost no RefSeq specific genes. Genes only detected with Gencode are longer and have on average more exons and transcripts.

(5) It was not clear how the kallisto TCCs were converted to transcript/gene counts, and at which point in the process "non-unique" alignments were discarded. Also, the authors write that one explanation for the low assignment rates of kallisto could be that "the 3'end of a gene alone often does not allow to distinguish different transcripts"; however, this is not necessary if only a gene level quantification is required.

We thank the reviewer for directing us to this issue. We now provide a more detailed explanation of the collapsing in Materials and Methods:

“To be comparable with other alignment methods, the counts per equivalence class were collapsed per gene. The counts of equivalence classes representing multiple genes were filtered out.” (page 13, line 333)

In Figure 1A, the lighter part of the bar represents all reads that were assembled into an equivalence class by kallisto. The darker part represents the proportion of reads for which the equivalence class could be uniquely assigned to one gene. Thus the removed reads were probably assembled to chimeric equivalence classes, that would map to multiple genes.

We now corrected the sentence: "One possible explanation is that the 3'end of a gene alone often ends up in chimeric equivalence classes." (page 12, line 259)

(6) What does "BWA alignment with assignment of reads to features" mean? The BWA commands in Table S3 do not contain any code to perform assignment of reads to features. Furthermore, the "bwa samse" command help page states that "Repetitive hits will be randomly chosen". Is this accounted for, or will a read that maps to multiple genes not be detected as such (which would affect the assignment rate, the number of detected genes as well as the variance)?

We have extended the explanation in the Material and Methods: (page 13, line 325)

"For transcriptome alignment, we downloaded transcriptome fasta files corresponding to the annotations listed above. We used BWA (v0.7.12) to align the scRNA-seq reads to these transcriptomes. We only removed reads that aligned equally well to transcripts of different genes as truly multi-mapped. The remaining reads were tallied up per gene. For UMI data, the reads were collapsed per gene by identity, similar to the strategy recommended in zUMIs."

In addition, we also provide the R and bash scripts implementing this approach in our github repository (<https://github.com/bvieth/scRNA-seq-pipelines>).

(7) In ref [2] (Fig. 6), for 64 cells or more, SmartSeq2 showed comparable power to the UMI protocols. Could you elaborate on what causes the low power for SmartSeq2 compared to the other protocols (including 10x Genomics) seen in Figures 2F/S7?

We believe that the lower power is due to the amplification noise introduced by the many initial PCR cycles necessary for scRNA-seq. UMIs allow to remove the amplification noise, this is visible when comparing the dispersion for the same mapper in Figure 2D.

Note that we are not repeating the analysis of Ziegenhain et al. 2017, in that we compare computational pipelines and not RNA-seq protocols. In Figure 2 we simply highlight key differences between protocols that will have repercussions on the remaining analysis. Therefore, we contrast first the number of genes detected using different protocols and mappers and then focus on the level of noise with which each detected gene is measured. In Ziegenhain et al. Figure 6 A, those two measures are combined in the power analysis, to honor the higher detection rate of Smart-Seq2.

(8) What is happening with the mean-dispersion relationship for the kallisto/SCRB-seq data (Figure S6)? It looks very different from the others (and from what is typically expected for RNA-seq data).

We think that the uptilt of the dispersion for high mean values for SCR-seq with kallisto could be a result from low number of highly expressed genes. There were only maximal 12 genes with a \log_2 mean normalised expression above 5 .

Because these are only so few genes, this uptick will not have a significant impact on our results.

(9) It is not clear whether the authors use 'normalisation factors' and 'size factors' to refer to the same concept. Typically, normalisation factors are used as correction factors and multiplied with the library sizes, while size factors incorporate also the library size differences. This is important, since some of the employed normalisation methods will return size factors, while others return normalisation factors. In addition, in the presence of differential expression, the correct size factors are not necessarily proportional to the library sizes (since normalisation factors are intended to avoid compositionality). It is not completely clear to me which values the authors compare with their RMSE evaluations.

Also, it is not clear from the text how the normalisation factors (or size factors) are used in the differential expression evaluation (the text says only "The derived normalisation scaling factors were then used in conjunction with the raw count matrices for DE-testing").

We thank the reviewer for pointing this out. Indeed, we always derived the normalisation factors for DE testing, just like Sonesson et al. 2018. For the RMSE evaluation, we centered and scaled the estimated size factors to compare them with the simulated size factors (see additional Supplementary Figure S10, referenced on page 15, line 387). This approach has already been used by Vallejos et al. 2017. We now specify to the appropriate factor - size factor and normalisation factor - throughout the text.

Figure S10: Illustration for RMSE evaluation of library size factors.

The expression of 10,000 genes over 768 cells (384 cells per group (red and cyan points) were simulated and 20% of the genes were differentially expressed following an asymmetric narrow gamma distribution. To compare the estimated library size factors with the simulated library size factors, the factors were centred and scaled. The root mean squared error (RMSE) of a robust linear regression represents the deviation between estimated and simulated size factors.

(10) The simulation model (specifically, the introduction of DE genes) is not sufficiently well described. For example, were effect sizes chosen independently of the expression level, and was the dropout probability adapted to the "new" expression level (after the introduction of differential expression)? Also, were the data simulated with different

library sizes and/or size factors between cells (if so, what was the range, and was it set to be similar to real data)?

We first draw the mean and dispersion, and then independently draw an effect size given as \log_2 fold changes. As we described in the original powsimR publication (Vieth et al., 2017), the mean shift alone is sufficient to explain also observed differences in the dropout rates and thus does not need to be explicitly altered. We have added additional explanation to the Material and Methods: “The magnitude of expression change is drawn from a narrow gamma distribution ($X \sim \text{Gamma}(\alpha = 1, \beta = 2)$) defining the \log_2 fold change which is added to the sampled mean expression.” (Page 14 Line 365)

To make the data realistic we draw a different size factor for each cell using a normal distribution (Figure 1B). We added this to the Material Methods: “To introduce slight variation in expression capture, we draw a different size factor for each cell from a narrow normal distribution (Figure 1B). This distribution fits the considered data sets well, irrespective of the applied library preparation method.” (Page 14 Line 358)

(11) How were the TPR vs FDR curves made in order to estimate the pAUC? With FDR on the x-axis (and TPR on the y-axis), there is no guarantee that this curve is a valid function curve, and thus the area under the curve is not readily interpretable.

We used the `calculate_performance()` function from the iCOBRA package to determine the true FDR cut-off and then adapted the `auc` option from `ROCR::performance` to calculate the pAUC based on rolling means (Soneson et al. 2016, Sing et al. 2005). We have included an additional supplementary figure to illustrate this procedure (Supplementary Figure S11) and reference it in the text (page 15, line 400).

Figure S11: Illustration for pAUC calculation used as an evaluation metric for performance. The expression of 10,000 genes estimated from SCR-seq data over 768 cells (384 cells per group were simulated and 20% of the genes were differentially expressed following an asymmetric narrow gamma distribution. The power to detect DE (TPR) versus observed FDR based on DE-testing with limma-trend using scan with clustering (**A**) or Median-Ratio (**B**) is plotted²⁰. The dashed line indicates the level at the which the observed stays below the nominal level of 10% which defines the right side boundary for the partial area under this curve²¹. The corresponding proportion and rates for a selection of nominal FDR thresholds are given in **C**, **D**. Since we do not want to punish conservative FDR control of methods, the test results using scan normalisation for example is extended to observed FDR of 10% whereas median ratio only ensures FDR control at a lower observed FDR.

Various comments

(12) Why were all raw sequencing reads cut and downsampled to 45 base long, 1M cDNA reads per cell?

That was done to make the protocols comparable since we had differences in read length design and sequencing depth in Ziegenhain et al. (2017) and we did not want to make our estimation of simulation parameters dependent on sequencing depth, mapping differences and gene detection limits.

(13) It could be made clearer that the alignment/assignment evaluation was done on real data, not the simulated ones.

We changed the Results section accordingly. Page 2, Line 54: “We first investigated how expression quantification is affected by different alignment methods using our selection of scRNA-seq experiments.”

(14) The statement in the Introduction that "All these goals have the first analysis steps leading to a filtered and normalised count matrix in common" is not quite true - many DE methods (and also increasingly methods for other types of analyses) work on the unnormalised counts and build library size normalisation etc. into the model (so a "normalised" matrix is never generated).

We changed this sentence to “All these goals have the first analysis steps in common in that they require expression counts or normalised counts.” Even though some methods perform these analysis steps internally, they are still necessary.

(15) MAST does not assume a zero-inflated negative binomial distribution, but a zero-inflated normal distribution of the log-transformed TPMs/CPMs.

We thank the reviewer for pointing this out. CPMs are no longer count data, however, they were generated from a count distribution. We changed this in the text:

“such as for example MASTs hurdle model which implicitly assumes that the CPM values were generated from zero inflated negative binomial count distribution. “

(16) The statement "in concordance with Andrews and Hemberg that in particular for sparse data such as 10X, SAVER imputation before normalisation improves performance" seems a bit overstated, given that Andrews and Hemberg's recommendation is that "SAVER is generally the safest method to use, but that all statistical tests, be the gene-gene correlations, cluster-specific marker genes, or differential expression, should be applied to un-imputed data." Also, it is important to note that SAVER is based on a negative binomial assumption, which as far as I can see was also the model used to generate the simulated data (in both studies).

We completely agree that no DE-analysis should be performed on imputed data. As in Andrews and Hemberg, we also only use an imputed count matrix to estimate the size factors, after that we return to the raw counts and only keep the size factors. We have added the following text (page 6, line 137): “Note, that we use the imputation or filtering only to obtain size factor estimates, that are then used together with the raw count matrix for DE-testing.”

Furthermore, in Vieth et al. (2017), we show that for UMI data the negative binomial is generally a good fit, only for non-UMI data such as Smart-seq2 a zero-inflated negative binomial distribution occasionally provides a better fit.

Therefore, for Smart-seq2, we used a zero-inflated negative binomial distribution for our simulations. We hopefully made this clearer in the Methods section: “Since we (Vieth et al., 2017) and others (Amrhein et al, 2019, Sonesson et al., 2019) have found previously, we assume that UMI counts follow a negative binomial distribution and only Smart-seq2 needs the inclusion of zero-inflation.” (page page 14, line 336). In our analysis SAVER did not perform differently for Smart-seq2 data (ZINB) as compared to UMI data (NB). However, imputation in general did not make a big difference overall.

(17) It should be made clearer early in the paper that the focus is on detection of differential gene expression between groups of cells. "Single-cell RNA-seq analysis pipelines" can refer to many other things, including trajectory inference, clustering, visualization and other types of differential analysis (e.g., differential abundance of cell types, or comparison of gene expression levels between samples from different conditions).

We have changed the Introduction. Page 1. line 7 "Here, we focus on these important first choices made in any scRNA-seq study, using DE-inference as performance read-out."

(18) How was the "frequency filtering" done?

We added a description of the gene dropout filtering to the Methods: "These data sets were then analysed by a nearly exhaustive number of combinations of four imputation strategies (scone, SAVER, DrImpute), gene dropout filtering (remove genes with more than 80% zero expression values) together with seven normalisation approaches (TMM, MR, Linnorm, Census, Linnorm, scran, SCnorm) with or without spike-ins, depending on library preparation protocol and method (Figure 1C)." (Page 14, Line 374)

(19) I would suggest that the authors give a precise definition of what they mean by the term "dropout rate" (i.e., total fraction of zeros, or zeros not explainable by the count model). For example, zingeR does not estimate the fraction of zeros, but a posterior probability of zeros coming from the count model.

We have changed dropout rate to gene dropout rate accordingly in the text and figure legends.

Reviewers' Comments:

Reviewer #1:

Remarks to the Author:

I would like to thank the authors for their detailed assessment of my review and for providing analytical support to all my concerns and questions. I do feel that the manuscript has advanced significantly.

Reviewer #2:

Remarks to the Author:

In the revised manuscript, the authors have addressed most of my original concerns. I have a few remaining comments:

(1) Regarding my previous comment about the kallisto quantifications, I am still surprised that the number of reads assigned by kallisto is so low - from Figure 2A it looks like most of the reads fall in equivalence classes with transcripts from multiple genes (whereas e.g. <https://genomebiology.biomedcentral.com/articles/10.1186/s13059-018-1419-z> excluded 12% of the reads with this filter). The authors now provide the code base that they used for their evaluations, and from the `run_kallisto_summarise.R` script (e.g., line 74), it seems that a read is considered "repeatedly mapped" if it is assigned to an equivalence class with multiple `_transcripts_` (not multiple genes). I am not sure whether those are the results that go into Figure 2A (I could not find that information in the provided code), but if so, that doesn't seem correct.

(2) With respect to the BWA alignment, I asked about the effect of "bwa samse" choosing randomly among repetitive hits. The authors replied that they only removed reads that aligned equally well to transcripts of different genes, but I would appreciate a confirmation that no further alignments were removed by the `bwa samse` command.

(3) The authors state that MAST assumes that "CPM values were generated from zero inflated negative binomial count distribution". However, MAST models the log-transformed normalized expression values, using a Gaussian (conditional on being expressed).

(4) In their rebuttal letter, the authors explain that the pAUCs were estimated from a graph with the FDR on the x-axis and the TPR on they-axis. However, while the TPR will increase with increasing p-value threshold, no such guarantee can be had for the observed FDRs. Thus, as I mentioned in my previous report, there is no guarantee that these curves actually represent function curves, and as such, I find it non-intuitive to calculate the pAUC as described by the authors. The typical way would be to instead consider the axis assignment of a precision-recall plot, which has recall (TPR) on the x-axis and precision (1-FDR) on the y-axis.

Now, we understand Reviewer 2's remaining comments better and thus can address them appropriately. Our clarifications can be found in green interleaved with the comments.

Reviewers' comments:

Reviewer #1 (Remarks to the Author):

I would like to thank the authors for their detailed assessment of my review and for providing analytical support to all my concerns and questions. I do feel that the manuscript has advanced significantly.

Thank you for your helpful review.

Reviewer #2 (Remarks to the Author):

In the revised manuscript, the authors have addressed most of my original concerns. I have a few remaining comments:

(1) Regarding my previous comment about the kallisto quantifications, I am still surprised that the number of reads assigned by kallisto is so low - from Figure 2A it looks like most of the reads fall in equivalence classes with transcripts from multiple genes (whereas e.g. <https://genomebiology.biomedcentral.com/articles/10.1186/s13059-018-1419-z> excluded 12% of the reads with this filter). The authors now provide the code base that they used for their evaluations, and from the `run_kallisto_summarise.R` script (e.g., line 74), it seems that a read is considered "repeatedly mapped" if it is assigned to an equivalence class with multiple `_transcripts_` (not multiple genes). I am not sure whether those are the results that go into Figure 2A (I could not find that information in the provided code), but if so, that doesn't seem correct.

Note that the data used in <https://genomebiology.biomedcentral.com/articles/10.1186/s13059-018-1419-z> is bulk RNAseq generated using a full length method. This means that each transcript has much better coverage than in single cell RNA-seq and unlike for the UMI protocols, it has little to no 3' bias. In agreement with this, we also see that kallisto works fine with data generated with the full length method Smart-seq2, but has problems with 3' UMI methods (Figure 2).

We are sorry for the confusion regarding the kallisto scripts. Indeed, we quantify the number of counts per equivalence class and associate it with the corresponding transcript and gene id. For UMI-collapsing we relied on the command line verbose output from kallisto that originally collapses UMIs per transcript. The table below summarises the command line verbose output from kallisto that we used for alignment and assignment rate quantification. In our R-script (code line 39-42 and 114-117 in `run_kallisto_summarise.R`), we use the UMI-counts per equivalence and the equivalence class assignments to gene ids from kallisto and summarise them per gene. In this process we remove all equivalence classes that are associated with more than one gene. This will mainly affect chimeric equivalence classes and also remove UMIs from the parts that could have been uniquely assigned. If the goal is comparing equivalence classes and not genes more UMIs could be retained.

We have included the necessary data to reproduce the manuscript figures on zenodo and the scripts on github.

Protocol	Annotation	Total reads	Pseudo-aligned	unique UMIs assigned	Gene Counts
SCRB-seq	gencode	93,000,000	59,408,133	10,542,781	2,851,688
SCRB-seq	vega	93,000,000	49,787,739	6,524,879	2,698,895
SCRB-seq	refseq	93,000,000	49,787,739	6,524,879	3,446,426
Drop-seq	gencode	79,000,000	52,846,626	6,075,570	1,372,872
Drop-seq	vega	79,000,000	48,694,027	5,661,164	1,329,032
Drop-seq	refseq	79,000,000	50,455,164	3,301,671	1,628,814
CEL-seq2	gencode	93,000,000	52,498,755	9,782,519	3,923,965
CEL-seq2	vega	93,000,000	42,974,187	8,684,980	3,702,633
CEL-seq2	refseq	93,000,000	52,842,757	8,663,640	5,587,987
10X Genomics - HGMM	gencode	19,384,329	14,990,238	12,122,418	3,744,817
10X Genomics - HGMM	vega	19,384,329	13,689,061	11,102,210	4,138,699
10X Genomics - HGMM	refseq	19,384,329	14,307,398	11,046,368	8,455,851
10X Genomics - PBMC	gencode	60,160,373	37,624,519	17,197,011	4,098,532
10X Genomics - PBMC	vega	60,160,373	35,747,539	16,690,036	4,305,855
10X Genomics - PBMC	refseq	60,160,373	30,727,071	10,995,591	4,877,732
Smart-seq2	gencode	92,000,000	49,772,178	-	20,981,948
Smart-seq2	vega	92,000,000	49,321,146	-	19,626,450
Smart-seq2	refseq	92,000,000	47,475,351	-	27,060,907

(2) With respect to the BWA alignment, I asked about the effect of "bwa samse" choosing randomly among repetitive hits. The authors replied that they only removed reads that aligned equally well to transcripts of different genes, but I would appreciate a confirmation that no further alignments were removed by the bwa samse command.

We first run `bwa aln` followed by `bwa samse` with default parameters (see `mapping.Rmd` line 91 ff.), which keeps per default up to 3 equally well mapping 'repetitive' hits. The transcript ids are then stored in the `XA:Z:` tag that we parse in the R-script `run_transcriptome_counts.R` line 58 ff. If those transcript-ids are

associated with the same gene-id, we keep the read as uniquely mapped and discard it otherwise.

(3) The authors state that MAST assumes that "CPM values were generated from zero inflated negative binomial count distribution". However, MAST models the log-transformed normalized expression values, using a Gaussian (conditional on being expressed).

We agree that the \log_2 expression values are modelled with a normal distribution in MAST. In order to fulfill this assumption, only genes that make the hurdle are considered. Thus, the original $\log_2(\text{CPM})$ distribution in MAST is normal plus the removed chunk of that is so close to zero that it did not make the hurdle. If, we would back transform such a distribution into the raw count space, it would be closest to a zero-inflated negative binomial. In brief, the use of a hurdle model alone implies a zero-inflation and it has been frequently reported that RNA-seq count data follow a negative binomial (Vieth et al. 2017; Grün et al. 2014). Therefore our statement is that MAST makes this assumption *implicitly*, does not contradict the fact that they use a normal.

(4) In their rebuttal letter, the authors explain that the pAUCs were estimated from a graph with the FDR on the x-axis and the TPR on the y-axis. However, while the TPR will increase with increasing p-value threshold, no such guarantee can be had for the observed FDRs. Thus, as I mentioned in my previous report, there is no guarantee that these curves actually represent function curves, and as such, I find it non-intuitive to calculate the pAUC as described by the authors. The typical way would be to instead consider the axis assignment of a precision-recall plot, which has recall (TPR) on the x-axis and precision ($1-\text{FDR}$) on the y-axis.

We agree that precision-recall plots (TPR vs. $1-\text{FDR}$) are a valid option to represent the trade-off in error rates of DE testing results (Saito & Rehmsmeier 2015). Nevertheless, plotting TPR vs FDR is an equivalent representation (Davis & Goadrich 2006) that has also been used in the context of DE analysis (e.g. (Mori et al. 2013)) and implemented

in a software tool for method evaluation and comparison (iCOBRA, (Soneson & Robinson 2016)).

We chose to limit TPR vs. FDR curve to a reasonable FDR range:

1. We believe that a nominal FDR level of 80% for example, is not of interest in most studies and therefore the partial AUC as a summary measure of performance up to 10% FDR is more informative.
2. When two competing curves cross, the AUC over the full range is no longer an unambiguous summary measure (Yousef 2013).

Indeed it can happen that multiple points of a TPR vs. FDR (or 1-FDR) curves have the same x-value, so that the curves would no longer be valid function curves. However, even if the curve cannot be defined as a function curve, valid approximations to compute the curve as well as the area under the curve exist. We chose a rolling mean as summary statistic to approximate the area under the curve, because it has been shown to be an unbiased estimator in the range of true positive DE genes that we simulated for our study (Boyd et al. 2013).

Boyd, K., Eng, K.H. & Page, C.D., 2013. Area under the Precision-Recall Curve: Point Estimates and Confidence Intervals. In *Machine Learning and Knowledge Discovery in Databases*. Springer Berlin Heidelberg, pp. 451–466. Available at: http://dx.doi.org/10.1007/978-3-642-40994-3_29.

Davis, J. & Goadrich, M., 2006. The Relationship Between Precision-Recall and ROC Curves. In *Proceedings of the 23rd International Conference on Machine Learning*. ICML '06. New York, NY, USA: ACM, pp. 233–240. Available at: <http://doi.acm.org/10.1145/1143844.1143874>.

Grün, D., Kester, L. & van Oudenaarden, A., 2014. Validation of noise models for single-cell transcriptomics. *Nature methods*, 11(6), pp.637–640. Available at: <http://dx.doi.org/10.1038/nmeth.2930>.

Mori, K. et al., 2013. Cancer outlier analysis based on mixture modeling of gene expression data. *Computational and mathematical methods in medicine*, 2013, p.693901. Available at: <http://dx.doi.org/10.1155/2013/693901>.

Saito, T. & Rehmsmeier, M., 2015. The precision-recall plot is more informative than the ROC plot when evaluating binary classifiers on imbalanced datasets. *PloS one*, 10(3), p.e0118432. Available at: <http://dx.doi.org/10.1371/journal.pone.0118432>.

Soneson, C. & Robinson, M.D., 2016. iCOBRA: open, reproducible, standardized and live method benchmarking. *Nature methods*, 13(4), p.283. Available at:

<http://dx.doi.org/10.1038/nmeth.3805>.

Vieth, B. et al., 2017. powsimR: power analysis for bulk and single cell RNA-seq experiments. *Bioinformatics* , 33(21), pp.3486–3488. Available at: <http://dx.doi.org/10.1093/bioinformatics/btx435>.

Yousef, W.A., 2013. Assessing classifiers in terms of the partial area under the ROC curve. *Computational statistics & data analysis*, 64, pp.51–70. Available at: <http://www.sciencedirect.com/science/article/pii/S0167947313000881>.

Reviewers' Comments:

Reviewer #2:

Remarks to the Author:

The authors have addressed my comments.

REVIEWERS' COMMENTS:

Reviewer #2 (Remarks to the Author):

The authors have addressed my comments.

We would like to thank again for the helpful reviews.